# GENERATIVE MODELING WITH PHASE STOCHASTIC BRIDGES

**Tianrong Chen**[1]*, **Jiatao Gu**[2], **Laurent Dinh**[2], **Evangelos A. Theodorou**[1]

**, Josh Susskind**[2]**, Shuangfei Zhai**[2]

[1]Georgia Tech, [2]Apple

{tianrong.chen, evangelos.theodorou}@gatech.edu, {jgu32,l_dinh,jsusskind,szhai}@apple.com

## ABSTRACT

We introduce a novel generative modeling framework grounded in phase space dynamics, taking inspiration from the principles underlying Critically damped Langevin Dynamics and Bridge Matching. Leveraging insights from Stochastic Optimal Control, we construct a more favorable path measure in the phase space that is highly advantageous for efficient sampling. A distinctive feature of our approach is the early-stage data prediction capability within the context of propagating generative Ordinary Differential Equations or Stochastic Differential Equations. This early prediction, enabled by the model's unique structural characteristics, sets the stage for more efficient data generation, leveraging additional velocity information along the trajectory. This innovation has spurred the exploration of a novel avenue for mitigating sampling complexity by quickly converging to realistic data samples. Our model yields comparable results in image generation and notably outperforms baseline methods, particularly when faced with a limited Number of Function Evaluations. Furthermore, our approach rivals the performance of diffusion models equipped with efficient sampling techniques, underscoring its potential in the realm of generative modeling.

## 1 INTRODUCTION

Diffusion Models (DMs;Song et al. (2020a); Ho et al. (2020)) constitute an instrumental technique in generative modeling, which formulate a particular Stochastic Differential Equation (SDE) linking the data distribution with a tractable prior distribution. Initially, a DM diffuses data towards the prior distribution via a predetermined linear SDE. In order to reverse the process, a neural network is used to approximate the score function which is analytically available. Subsequently, the approximated score is utilized to conduct time reversal (Anderson, 1982; Haussmann & Pardoux, 1986) of this diffusion process, ultimately generating data. Recently, the Critical-damped Langevin Dynamics (CLD;Dockhorn et al. (2021)) extends the SDE framework of DM into phase space (whereas DMs operate in the position space) by introducing an auxiliary velocity variable, which is defined by tractable Gaussian distributions at the initial and terminal time steps. This augmentation induces a trajectory in position space exhibiting enhanced smoothness, as stochasticity is solely introduced into the velocity space. The distinctive structure of CLD is shown to enhance the empirical performance and sample efficiency. However, despite the success of CLD, inefficient sampling still persists due to unnecessary curvature of the dynamics (Fig.1) as it has to converge to equilibrium for sampling from the tractable prior.

The remarkable accomplishments of DM have also catalyzed recent advancements in generative modeling, leading to the development of Bridge Matching (BM;(Peluchetti, 2021; Liu et al., 2022; 2023)) and Flow Matching (FM;models(Lipman et al., 2022)). These models leverage dynamic transport maps underpinned by the utilization of SDEs or ODEs. Unlike DM, Bridge and Flow Matching relaxes the reliance on a forward diffusion process with an asymptotic convergence to a prior distribution over an infinite time horizon. Moreover, they exhibit a heightened degree of versatility, enabling the construction of transport maps between two arbitrary distributions by drawing

---

*work done while Tianrong Chen is an intern at Apple

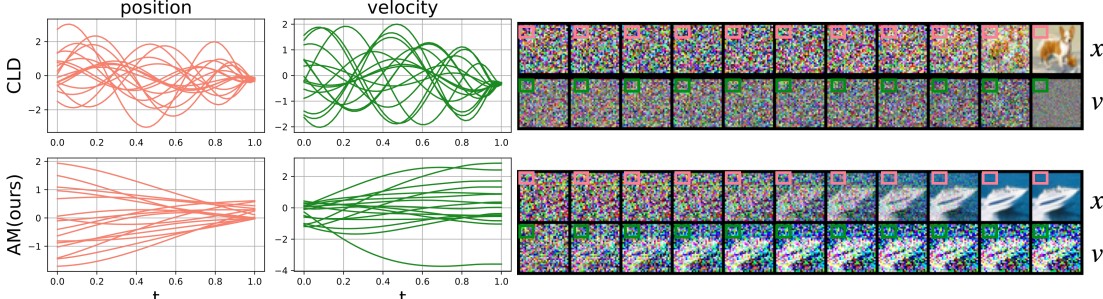

Figure 1: The pixel-wise trajectories comparison with CLD(Dockhorn et al., 2021). Left figures correspond to the trajectories over time w.r.t random sampled 16 pixels, for position and velocity. Our model is able to learn straighter trajectories which is beneficial for reducing sampling complexity.

upon insights from domains such as optimal transport (Pooladian et al., 2023), normalizing flow (Tong et al., 2023b), and optimal control (Liu et al., 2023).

In this paper, we focus on enhancing the sample efficiency of velocity based generative modeling (eg, CLD) by utilizing the Stochastic Optimal Control (SOC) theory. Specifically, we leverage the outcomes of stochastic bridge within the context of linear momentum systems (Chen & Georgiou, 2015) to construct a path measure bridging the data and prior distribution. The resulting path exhibits a more straight position and velocity trajectory compared to CLD (fig.1), making it more amenable to efficient sampling. Within the broader landscape of dynamic generative modeling (ie, ODE/SDE based generative models), data point can often be represented as linear combinations of scaled intermediate data of dynamics and Gaussian noise. In our work, we re-establish this property, enabling the estimation of target data points by leveraging both *state and velocity* information. In the case of DM and FM, the estimation of target data is exclusively reliant on position information, whereas our method incorporates the additional dimension of velocity data, enhancing the precision and comprehensiveness of our estimations. It is also worth noting that our model exhibits the capacity to generate high fidelity images at early time steps (fig.2). In addition, we propose a sampling technique which demonstrates competitive results with small Number of Function Evaluations (NFEs), eg, 5 to 10. Table.1 demonstrates the design differences among aforementioned models. In summary, our paper presents the following contributions:

1. We propose Acceleration Generative Modeling (AGM) which is built on the SOC theory, enabling the favorable trajectories for efficient sampling over 2nd-order momentum dynamics generative modeling such as CLD.

2. As a result of AGM structural characteristics, it becomes possible to estimate a realistic data point at an early time point, a concept we refer to as sampling-hop. This approach not only yields a significant reduction in sampling complexity but also offers a novel perspective on accelerating the sampling in generative modeling by leveraging additional information from the dynamics.

3. We achieve competitive results compared to DM approaches equipped with specifically designed fast sampling techniques on image datasets, particularly in small NFE settings.

## 2 PRELIMINARY

**Notation**: Let $\mathbf{x}_t \in \mathbb{R}^d$ and $\mathbf{v}_t \in \mathbb{R}^d$ denote the $d$-dimensional position and velocity variable of a particle $\mathbf{m}_t = [\mathbf{x}_t, \mathbf{v}_t]^\top \in \mathbb{R}^{2d}$ at time $t$. We denote the discretized time series as $0 \leq t_0 < ...t_n < t_N < 1$. The Wiener Process is denoted as $\mathbf{w}_t$. The identity matrix is denoted as $\mathbf{I}_d \in \mathbb{R}^{d \times d}$. We define $\mathbf{\Sigma}_t$ as the covariance matrix of $\mathbf{x}_t$ and $\mathbf{v}_t$ at time step $t$.

### 2.1 DYNAMICAL GENERATIVE MODELING

The generative modeling approaches rooted in dynamical systems, including ODE and SDE, have garnered significant attention. Here, we present three noteworthy dynamical generative models: Diffusion Model (DM), Flow Matching (FM) and Bridge Matching (BM).

Table 1: Comparison between models in terms of boundary distributions $p_0$ and $p_1$. Our AGM generalizes DM beyond Gaussian priors to phase space, similar to CLD. However, unlike CLD, AGM does not need to converge to the Gaussian at equilibrium which causes curved trajectory(see Fig.1), instead, velocity distribution will be the convolution of data distribution with Gaussian.

| Models | DM/FM | CLD | AGM(ours) |
|---|---|---|---|
| $p_0(\cdot)$ | $p_{\text{data}}(x)$ | $p_{\text{data}}(x) \times \mathcal{N}(\mathbf{0}, \boldsymbol{I}_d)$ | $\mathcal{N}(\mathbf{0}, \boldsymbol{\Sigma}_0 \times \boldsymbol{I}_{2d})$ |
| $p_1(\cdot)$ | $\mathcal{N}(\mathbf{0}, \boldsymbol{I}_d)$ | $\mathcal{N}(\mathbf{0}, \boldsymbol{I}_d) \times \mathcal{N}(\mathbf{0}, \boldsymbol{I}_d)$ | $p_{\text{data}}(x) \times p_{\text{data}}(x) * \mathcal{N}(\mathbf{0}, \boldsymbol{\Sigma}_1 \otimes \boldsymbol{I}_{2d})$ |

**Diffusion Model:** In the framework of DM, given $\mathbf{x}_0$ drawn from a data distribution $p_{\text{data}}$, the model proceeds to construct a SDE,

$$d\mathbf{x}_t = f_t(\mathbf{x}_t)dt + g(t)d\mathbf{w}_t \quad \mathbf{x}_0 \sim p_{\text{data}}(\mathbf{x}) \tag{1}$$

whose terminal distributions at $t = 1$ approach an approximate Gaussian, i.e. $\mathbf{x}_1 \sim \mathcal{N}(\mathbf{0}, \boldsymbol{I}_d)$. This accomplishment is realized through the careful selection of the diffusion coefficient $g_t$ and the base drift $f_t(\mathbf{x}_t)$. It is noteworthy that the time-reversal (Anderson, 1982) of (1) results in another SDE:

$$d\mathbf{x}_t = \left[ f_t(\mathbf{x}_t) - g_t^2 \nabla_{\mathbf{x}} \log p(\mathbf{x}_t, t) \right] dt + g(t)d\mathbf{w}_t, \quad \mathbf{x}_1 \sim \mathcal{N}(\mathbf{0}, \mathbf{I}_d) \tag{2}$$

where $p(\cdot, t)$ is the marginal density of (1) at time $t$ and $\nabla_{\mathbf{x}} \log p_t$ is known as the score function. SDE (2) can be regarded as the time-reversal of (1) in such a manner that the path-wise measure is almost surely equivalent to the one induced by (1). As a consequence, these two SDEs share identical marginal over time. In practice, it is feasible to analytically sample $\mathbf{x}_t$ given $t$ and $\mathbf{x}_0$. Additionally, we can leverage a neural network to learn the score function by regressing scaled Stein Score $\mathbb{E}_{\mathbf{x}_t, t} \| \mathbf{s}_t^\theta(\mathbf{x}_t, t; \theta) - \nabla_{\mathbf{x}} \log p(\mathbf{x}_t, t | \mathbf{x}_0) \|_2^2$ for the purpose of propagating (2). This learned score can then be integrated into the solution of the aforementioned SDE(2) to simulate the generation of data that adheres to the target data distribution from the prior distribution. Meanwhile, (2) also corresponds to an ODE which shares the same path-wise measure:

$$d\mathbf{x}_t = \left[ f_t(\mathbf{x}_t) - \frac{1}{2} g_t^2 \nabla_{\mathbf{x}} \log p(\mathbf{x}_t, t) \right] dt, \quad \mathbf{x}_1 \sim \mathcal{N}(\mathbf{0}, \mathbf{I}_d) \tag{3}$$

which motivates the popular sampler introduced in (Zhang & Chen, 2022; Zhang et al., 2022; Bao et al., 2022) to solve the ODE (2) efficiently.

**Bridge Matching and Flow Matching:** An alternative approach to exploring the time-reversal of a forward noising process involves the concept of 'building bridges' between two distinct distributions $p_0(\cdot)$ and $p_1(\cdot)$. This method entails the learning of a mimicking diffusion process, commonly referred to as bridge matching, as elucidated in previous works (Peluchetti, 2021; Shi et al., 2022). Here we consider the SDE in the form of:

$$d\mathbf{x}_t = \mathbf{v}_t(\mathbf{x}, t)dt + g_t d\mathbf{w}_t \quad s.t. \quad (x_0, x_1) \sim \Pi_{0,1}(\mathbf{x}_0, \mathbf{x}_1) := p_0 \times p_1 \tag{4}$$

which is pinned down at an initial and terminal point $x_0, x_1$ which are independently samples from predefined $p_0$ and $p_1$. This is commonly known as the reciprocal projection of $x_0$ and $x_1$ in the literature (Shi et al., 2023; Peluchetti, 2023; Liu et al., 2022; Léonard et al., 2014). The construction of such SDE is accomplished by meticulous design of $\mathbf{v}_t$. A widely adopted choice for $\mathbf{v}_t$ is $\mathbf{v}_t := (\mathbf{x}_1 - \mathbf{x}_t)/(1 - t)$, which induces the well-known Brownian Bridge (Liu et al., 2023; Somnath et al., 2023). Similar to the approach in DM and owing to the linear structure of the dynamics, one can efficiently estimate this drift by employing a neural network parameterized by weights $\theta$ for regression on: $\mathbb{E}_{\mathbf{x}_t, t} \| \mathbf{v}_t^\theta(\mathbf{x}_t, t; \theta) - \mathbf{v}_t(\mathbf{x}_t, t) \|_2^2$ given $\mathbf{x}_1$ and $t$. As extensively discussed in previous studies (Liu et al., 2023; Shi et al., 2022), this bridge matching framework takes on the characteristics of FM (Lipman et al., 2022) when the diffusion coefficient $g_t$ tends to zero.

**Remark 1.** *The practice of constraining a stochastic process to specific initial and terminal conditions is a well-established setup in SOC. For a gentle introduction of it's connection with Brownian Bridge, Schrödinger Bridge please see Appendix.C. From this perspective, one can derive Brownian Bridge, as elaborated in Appendix.D.1 for comprehensive elucidation. It is imperative to note that the SOC framework will serve as the fundamental basis upon which we will develop our algorithm.*

## 3   ACCELERATION GENERATIVE MODEL

We apply SOC to characterize the twisted trajectory of momentum dynamics induced by CLD(Dockhorn et al., 2021). It becomes evident that the mechanisms encompassing flow matching, diffusion modeling, and Bridge matching collectively facilitate the construction of an estimated target data point, denoted as $\mathbf{x}_1$, by utilizing the intermediate state of the dynamics, $\mathbf{x}_t$. Our additional objective is to expedite the estimation of a plausible $\mathbf{x}_1$ by incorporating additional dynamics-related information, such as velocity, thereby curtailing the requisite time integration.

In this section, we introduce the proposed method, termed as the Acceleration Generative Model (AGM), rooted in SOC theory. Building upon (Chen & Georgiou, 2015), we extend the framework by incorporating a time-varying diffusion coefficient and accommodating arbitrary boundary conditions, ultimately arriving at an analytical solution suited for the generative modeling. We demonstrate its efficacy in rectifying the trajectory of CLD, concurrently showcasing its aptitude for accurately estimating the target data at an early timestep $t_i$, thereby enabling expeditious sampling.

As suggested by BM approach, there is a necessity to formulate a trajectory that bridges the two data points sampled from $p_0$ and $p_1$ respectively. Desirably, the intermediate trajectory should exhibit optimal characteristics that facilitate smoothness and linearity. This is essential for the ease of simulating the dynamics system to obtain the solution. In our endeavor to tackle this challenge and enhance the estimation of the data point $\mathbf{x}_1$ by incorporating velocity components, we encapsulate the problem within a SOC framework, specifically formulated in the phase space which reads:

**Definition 2** (Stochastic Bridge problem of linear momentum system (Chen & Georgiou, 2015))**.**

$$\min_{\mathbf{a}_t} \int_\tau^1 \|\mathbf{a}_t\|_2^2 \mathrm{d}t + (\mathbf{m}_1 - m_1)^\mathsf{T} \mathbf{R} (\mathbf{m}_1 - m_1) \quad s.t \underbrace{\begin{bmatrix} \mathrm{d}\mathbf{x}_t \\ \mathrm{d}\mathbf{v}_t \end{bmatrix}}_{\mathrm{d}\mathbf{m}_t} = \underbrace{\begin{bmatrix} \mathbf{v}_t \\ \mathbf{a}_t(\mathbf{x}_t, \mathbf{v}_t, t) \end{bmatrix}}_{\mathbf{f}(\mathbf{m},t)} \mathrm{d}t + \underbrace{\begin{bmatrix} \mathbf{0} & \mathbf{0} \\ \mathbf{0} & g_t \end{bmatrix}}_{\mathbf{g}_t} \mathrm{d}\mathbf{w}_t,$$

$$\mathbf{m}_\tau := \begin{bmatrix} \mathbf{x}_\tau \\ \mathbf{v}_\tau \end{bmatrix} = \begin{bmatrix} x_\tau \\ v_\tau \end{bmatrix}, \quad \mathbf{R} = \begin{bmatrix} \mathbf{r} & \mathbf{0} \\ \mathbf{0} & \mathbf{r} \end{bmatrix} \otimes \boldsymbol{I}_d, \quad x_1 \sim p_{\mathrm{data}}. \tag{5}$$

In this context, the matrix $\mathbf{R}$ is recognized as the terminal cost matrix, serving to assess the proximity between the propagated $\mathbf{m}_1$ and the ground truth $m_1$ at the terminal time $t = 1$. As the parameter $\mathbf{r}$ approaches positive infinity, the trajectory converges toward the state $x_1$, prompting a transition to constrained dynamics wherein the system becomes constrained by two predetermined boundaries, namely $m_0$ and $m_1$. This configuration aligns seamlessly with the principles of constructing a feasible bridge, as advocated by the tenets of BM. It is worth noting that this interpolation approach essentially represents a natural extension (Chen & Georgiou, 2015) of the well-established concept of the Brownian Bridge (Revuz & Yor, 2013), which has been employed in trajectory inference (Somnath et al., 2023; Tong et al., 2023a) and image inpainting tasks (Liu et al., 2023) and its connection with Diffusion has been discussed in Liu et al. (2023). Indeed, it is evident that the target velocity lacks a precise definition within this problem, allowing for flexibility in the design space for our approach. To address this, we opt for the linear interpolation of the intermediate point and the target point, represented as $\mathbf{v}_1 = (\mathbf{x}_1 - \mathbf{x}_t)/(1 - t)$, as the chosen terminal velocity, which also is the optimal control in the original space (see Appendix..D.1). This choice is made due to its ability to construct a trajectory characterized by straightness. Conceptually, the acceleration $\mathbf{a}_t$ continually guides the dynamics towards the linear interpolation of the two data points, serving to mitigate the impact of introduced stochasticity. In contrast to previous bridge matching frameworks, the velocity's boundary condition in our approach *varies over time* since it depends on the state $\mathbf{x}_t$ and $t$. The velocity variable serves solely as an auxiliary component aimed at straightening the trajectories. Regarding this SOC problem formulation, the solution is,

**Proposition 3** (Phase Space Brownian Bridge)**.** *When* $\mathbf{r} \to +\infty$*, The solution w.r.t optimization problem 5 is,*

$$\mathbf{a}^*(\mathbf{m}_t, t) = g_t^2 P_{11} \left( \frac{\mathbf{x}_1 - \mathbf{x}_t}{1 - t} - \mathbf{v}_t \right) \quad where: \quad P_{11} = \frac{-4}{g_t^2(t - 1)}. \tag{6}$$

*Proof.* Please see Appendix.D.2.  □

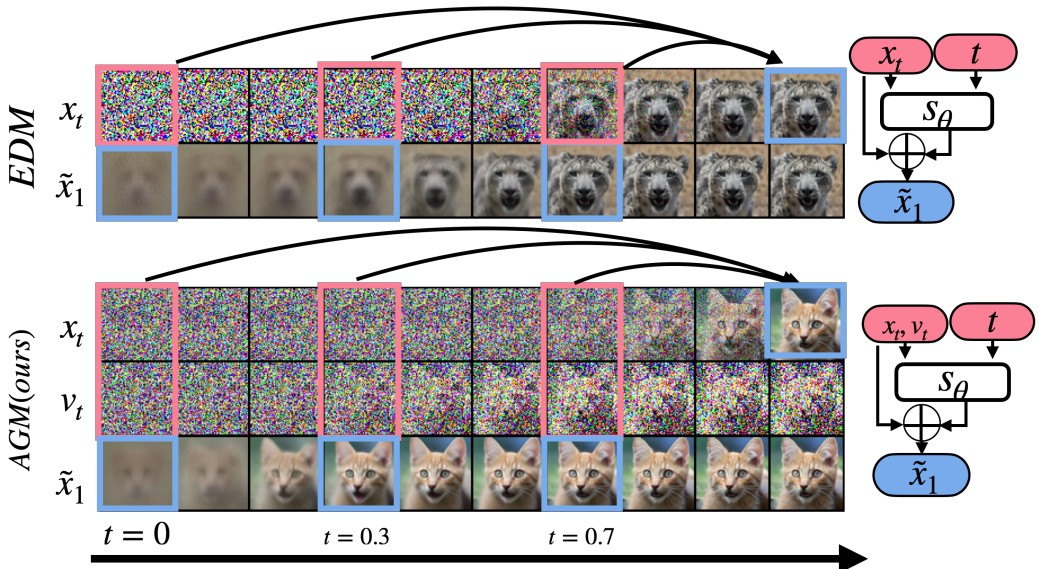

Figure 2: Data estimation comparison with EDM (Karras et al., 2022). When the network is endowed with supplementary velocity, AGM gains the capacity to estimate the target data point during the early stages of the trajectory. One can use estimated image $\tilde{\mathbf{x}}_1$ at $t_i < t_N$ as generated results and allocated more NFE between time $[0, t_i]$ which results to smaller discretization error.

**Remark 4.** *$P_{11}$ denotes the second diagonal component in the matrix $P_t$, a solution derived from the Lyapunov equation (see Lemma.9), serving as an implicit representation of the optimality of the control. This value is dependent upon the uncontrolled dynamics, where $\mathbf{a}_t$ is set to the zero vector in (5), and will vary accordingly when uncontrolled dynamics change.*

## 3.1 TRAINING

By plugging the optimal control (6) back to the dynamics (5), we can obtain the desired SDE. As been suggested by (Song et al., 2020b; Dockhorn et al., 2021), such SDE has a corresponding probablistic ODE which shares the same marginal over time in which the drift term will have an additional score term $\nabla_{\mathbf{v}} \log p(\mathbf{m}_t, t)$. Here we summarize the force term for SDE and ODE as:

$$\begin{bmatrix} \mathrm{d}\mathbf{x}_t \\ \mathrm{d}\mathbf{v}_t \end{bmatrix} = \begin{bmatrix} \mathbf{v}_t \\ \mathbf{F}_t \end{bmatrix} \mathrm{d}t + \begin{bmatrix} \mathbf{0} & \mathbf{0} \\ \mathbf{0} & h_t \end{bmatrix} \mathrm{d}\mathbf{w}_t \quad \text{s.t} \quad \mathbf{m}_0 := \begin{bmatrix} \mathbf{x}_0 \\ \mathbf{v}_0 \end{bmatrix} \sim \mathcal{N}(\boldsymbol{\mu}_0, \boldsymbol{\Sigma}_0),$$

Bridge Matching SDE : $\mathbf{F}_t := \mathbf{F}_t^b(\mathbf{m}_t, t) \equiv \mathbf{a}_t^*(\mathbf{m}_t, t), \qquad h(t) := g(t),$ (7)

Probablistic ODE : $\mathbf{F}_t := \mathbf{F}_t^p(\mathbf{m}_t, t) \equiv \mathbf{a}_t^*(\mathbf{m}_t, t) - \frac{1}{2}g_t^2 \nabla_{\mathbf{v}} \log p(\mathbf{m}, t), \quad h(t) := 0.$

Henceforth, we refer to the dynamics associated with the Bridge Matching SDE as AGM-SDE, and its corresponding ODE counterpart as AGM-ODE. Meanwhile, the linearity of the system implies the intermediate state $\mathbf{m}_t$ and the close form solution of score term are analytically available. In particular, the mean $\boldsymbol{\mu}_t$ and covariance matrix $\boldsymbol{\Sigma}_t$ of the intermediate marginal $p_t(\mathbf{m}_t|\mathbf{x}_1) = \mathcal{N}(\boldsymbol{\mu}_t, \boldsymbol{\Sigma}_t)$ of such a system can be analytically computed with $\boldsymbol{\Sigma}_t = \begin{bmatrix} \Sigma_t^{xx} & \Sigma_t^{xv} \\ \Sigma_t^{xv} & \Sigma_t^{vv} \end{bmatrix} \otimes \mathbf{I}_d$, and $\boldsymbol{\mu}_t = \begin{bmatrix} \mu_t^x \\ \mu_t^v \end{bmatrix}$, provided we have the boundary conditions $\boldsymbol{\mu}_0$ and $\boldsymbol{\Sigma}_0$ in place, as outlined in Särkkä & Solin (2019). Please see Appendix.D.3 for detail. In order to sample from such multi-variant Gaussian, one need to decompose the covariance matrix by Cholesky decomposition, and $\mathbf{m}_t$ is reparamertized as:

$$\mathbf{m}_t = \boldsymbol{\mu}_t + \mathbf{L}_t \boldsymbol{\epsilon} = \boldsymbol{\mu}_t + \begin{bmatrix} L_t^{xx} \boldsymbol{\epsilon}_0 \\ L_t^{xv} \boldsymbol{\epsilon}_0 + L_t^{vv} \boldsymbol{\epsilon}_1 \end{bmatrix}, \nabla_{\mathbf{v}} \log p_t := -\ell_t \boldsymbol{\epsilon}_1 \tag{8}$$

where $\boldsymbol{\Sigma}_t = \mathbf{L}_t \mathbf{L}_t^\mathsf{T}, \boldsymbol{\epsilon} = \begin{bmatrix} \boldsymbol{\epsilon}_0 \\ \boldsymbol{\epsilon}_1 \end{bmatrix} \sim \mathcal{N}(\mathbf{0}, \mathbf{I}_{2d})$ and $\ell_t = \sqrt{\frac{\Sigma_t^{xx}}{\Sigma_t^{xx}\Sigma_t^{vv} - (\Sigma_t^{xv})^2}}$.

**Parameterization**: The Force term can be represented as a composite of the data point and Gaussian noise. Specifically,

$$\mathbf{a}^*(\mathbf{m}_t, t) = 4\mathbf{x}_1(1-t)^2 - g_t^2 P_{11} \left[ \left( \frac{L_t^{xx}}{1-t} + L_t^{xv} \right) \boldsymbol{\epsilon}_0 + L_t^{vv} \boldsymbol{\epsilon}_1 \right]. \tag{9}$$

We express the force term as $\mathbf{F}_t^\theta = \mathbf{s}_t^\theta \cdot \mathbf{z}_t$. Here, $\mathbf{z}_t$ assumes the role of regulating the output of the network $\mathbf{s}_t^\theta$, ensuring that the variance of the network output is normalized to unity. For the detailed formulation of the normalizer $\mathbf{z}_t$, please refer to Appendix.D.8. In a manner similar to the BM approach, one can formulate the objective function for regressing the force term as follows:

$$\min_\theta \mathbb{E}_{t\in[0,1]} \mathbb{E}_{\mathbf{x}_1 \sim p_{\text{data}}} \mathbb{E}_{\mathbf{m}_t \sim p_t(\mathbf{m}_t|\mathbf{x}_1)} \lambda(t) \left[ \|\mathbf{F}_t^\theta(\mathbf{m}_t, t; \theta) - \mathbf{F}_t(\mathbf{m}_t, t)\|_2^2 \right] \tag{10}$$

Where $\lambda(t)$ is known as the reweight of the objective function across the time horizon. We defer the derivation of $\ell_t$ and the presentation of $\mathbf{L}_t$, $\lambda(t)$ and $\mathbf{a}_t$ in Appendix.D.

## 3.2 SAMPLING FROM AGM

Once the paramterized force term $\mathbf{F}_t^\theta$ is trained, we are ready to simulate the dynamics to generate the samples by plugging it back to the dynamics (7). One can use any type of SDE or ODE sampler to propagate the learnt system. Here we list our choice of sampler for AGM-SDE and AGM-ODE.

**Stochastic Sampler:** To simulate the SDE, prior works are majorly relying on Euler-Maruyama(EM) (Kloeden et al., 1992) and related methods. We adopt the Symmetric Splitting Sampler(SSS) from Dockhorn et al. (2021) in our AGM-SDE. This selection is based on the compelling performance it offers when dealing with momentum systems.

**Deterministic Sampler:** It is imperative to acknowledge that this system is inherently underactuated because the force term is exclusively injected into the velocity component, while velocity serves as the driving factor for the position—a variable of primary interest in generative modeling context. More specifically, at time step $t_i$, the impact of force does not immediately manifest in the position but rather takes effect at a subsequent time step, denoted as $t_{i+1}$ after discretizing time horizon. At time $t_0$, it becomes undesirable to propagate the state $\mathbf{x}_0$ using an initially uncontrolled velocity over an extended time interval $\delta_0$. The presence of this delay phenomenon can also exert an influence when the time interval $\delta_t$ is large, thereby impeding our ability to reduce the NFE during sampling. We propose the adoption of an Exponential Integrator (EI) approach, as elaborated in Zhang & Chen (2022). Empirical evidence suggests that this method aligns well with our model. We provide an illustrative example of how the AGM-ODE, in conjunction with the EI technique, can be employed to inject the learnt network into both velocity and position channels simultaneously:

$$\begin{bmatrix} \mathbf{x}_{t_{i+1}} \\ \mathbf{v}_{t_{i+1}} \end{bmatrix} = \Phi(t_{i+1}, t_i) \begin{bmatrix} \mathbf{x}_t \\ \mathbf{v}_t \end{bmatrix} + \sum_{j=0}^{w} \begin{bmatrix} \int_{t_i}^{t_{i+1}} (t_{i+1}-\tau) \mathbf{z}_\tau \cdot \mathbf{M}_{i,j}(\tau)\mathrm{d}\tau \cdot \mathbf{s}_t^\theta(\mathbf{m}_{t_{i-j}}, t_{i-j})) \\ \int_{t_i}^{t_{i+1}} \mathbf{z}_\tau \cdot \mathbf{M}_{i,j}(\tau)\mathrm{d}\tau \cdot \mathbf{s}_t^\theta(\mathbf{m}_{t_{i-j}}, t_{i-j}) \end{bmatrix}$$

$$\text{Where } \mathbf{M}_{i,j}(\tau) = \prod_{k\neq j} \left( \frac{\tau - t_{i-k}}{t_{i-j} - t_{i-k}} \right), \quad \text{and} \quad \Phi(t, s) = \begin{bmatrix} 1 & t-s \\ 0 & 1 \end{bmatrix}. \tag{11}$$

In Eq.11, $\Phi(s, t)$ denotes the transition matrix for our system, while $\mathbf{M}_{i,j}(\tau)$ represents the $w-$order multistep coefficient (Hochbruck & Ostermann, 2010). For a comprehensive derivation of these terms, please refer to Appendix.D.9. It is worth noting that the mapping of $\mathbf{s}_\theta$ into both the position and velocity channels significantly emulates the errors introduced by discretization delays. **Sampling-hop:** In the context of CLD (Dockhorn et al., 2021), their focus is on estimating the score function w.r.t. velocity, which essentially corresponds to estimating scaled $\boldsymbol{\epsilon}_1$ in our notation. However, relying solely on the aforementioned information is not sufficient for estimating the data point $\mathbf{x}_1$. Additional knowledge regarding $\boldsymbol{\epsilon}_0$ is also required in order to perform such estimation. In our case, the training objective implicitly includes both $\boldsymbol{\epsilon}_0$ and $\boldsymbol{\epsilon}_1$ (see eq.9), hence one can manage to recover $\mathbf{x}_1$ by Proposition.5. Remarkably, our observations have unveiled that when the network is equipped with additional velocity information, it acquires the capability to estimate the target data point during the early stages of the trajectory, as illustrated in fig.2. This estimation can be seamlessly integrated into AGM-SDE and AGM-ODE and we name it sampling-hop. Specifically,

**Proposition 5** (Sampling-Hop). *Given the state, velocity and trained force term $\mathbf{F}_t^\theta$ at time step $t$ in sampling phase, The estimated data point $\tilde{\mathbf{x}}_1$ can be represented as*

$$\tilde{\mathbf{x}}_1^{SDE} = \frac{(1-t)(\mathbf{F}_t^\theta + \mathbf{v}_t)}{g_t^2 P_{11}} + \mathbf{x}_t, \; or \; \tilde{\mathbf{x}}_1^{ODE} = \frac{\mathbf{F}_t^\theta + g_t^2 P_{11}(\alpha_t \mathbf{x}_t + \beta_t \mathbf{v}_t)}{4(t-1)^2 + g_t^2 P_{11}(\alpha_t \mu_t^x + \beta_t \mu_t^v)} \quad (12)$$

*for AGM-SDE and AGM-ODE dynamics respectively, and $\beta_t = L_t^{vv} + \frac{1}{2P_{11}}, \alpha_t = \frac{(\frac{L_t^{xx}}{1-t} + L_t^{xv}) - \beta_t L_t^{xv}}{L_t^{xx}}$.*

*Proof.* See Appendix.D.10 □

This property empowers us to allocate the NFE budget selectively within the time interval $t \in [0, t_i]$, where $t_i < t_N$, effectively reducing the discretization error while maintaining the sampling quality. This insight paves the way for efficient low NFE sampling strategies later. Here we summarized the training and sampling procedure of our method in Algorithm.1 and Algorithm.2 respectively.

---

**Algorithm 1** Training

1: **Input:** data distribution $p_{\text{data}}(\cdot)$
2: **while** not converge **do**
3:     $t \sim \mathcal{U}([0,1])$, $\mathbf{x}_1 \sim p_{\text{data}}(\mathbf{x}_1)$
4:     Compute mean and covariance $\boldsymbol{\mu}_t$ and $\boldsymbol{\Sigma}_t$. (Appendix.D.3)
5:     Sample $\mathbf{m}_t = \boldsymbol{\mu}_t + \mathbf{L}_t \boldsymbol{\epsilon}$.(eq.8)
6:     Compute target $\mathbf{F}_t$ (eq.7) using optimal acceleration (eq.9)
7:     Compute loss $\mathbb{E}\left[\lambda \|\mathbf{F}_t^\theta - \mathbf{F}_t\|_2^2\right]$(eq.10).
8:     Take gradient descent with respect to $\mathbf{F}_t^\theta(\mathbf{m}_t, t; \theta)$.
9: **end while**

---

**Algorithm 2** Sampling

1: **Input:** trained $\mathbf{F}(\cdot, \cdot; \theta)$, discretized time step $[t_0, \cdots, t_i]$, Choose the **sampler** from [SSS(SDE), EI(ODE)]. Choose prior mean and covariance $\boldsymbol{\mu}_0, \boldsymbol{\Sigma}_0$.
2: Sample $\mathbf{m}_0 \sim p_0(\mathbf{m}; \boldsymbol{\mu}_0, \boldsymbol{\Sigma}_0)$.
3: **for** n = 0 to $i$ **do**
4:     estimate $\mathbf{F}_{t_n}^\theta(\mathbf{m}_{t_n}, t_n)$
5:     $\mathbf{m}_{t_{n+1}} = \textbf{Sampler}(\mathbf{m}_{t_n}, F_{t_n}^\theta, t_n)$
6:     reconstruct $\hat{\mathbf{x}}_1$ using Proposition.5.
7: **end for**
8: **Return** $\hat{\mathbf{x}}_1$

---

## 4 EXPERIMENTAL RESULTS

**Architectures and Hyperparameters:** We parameterize $\mathbf{s}_t^\theta(\cdot, \cdot; \theta)$ using modified NCSN++ model as provided in Karras et al. (2022). We employ six input channels, accounting for both position and velocity variables, as opposed to the standard three channels used in the CIFAR-10 (Krizhevsky et al., 2009), AFHQv2 (Choi et al., 2020) and ImageNet (Deng et al., 2009) which leads to a negligible increase of network parameters. For the purpose of comparison with CLD in the toy dataset, we adopt the same ResNet-based architecture utilized in CLD. Throughout all of our experiments, we maintain a monotonically decreasing diffusion coefficient, given by $g(t) = 3(1-t)$. For the detailed experimental setup, please refer further to Appendix.E.

**Evaluation**: To assess the performance and the sampling speed of various algorithms, we employ the Fréchet Inception Distance score (FID;Heusel et al. (2017)) and the Number of Function Evaluations (NFE) as our metrics. For FID evaluation, we utilize reference statistics of all datasets obtained from EDM (Karras et al., 2022) and use 50k generated samples to evaluate. Additionally, we re-evaluate the FID of CLD and EDM using the same reference statistics to ensure consistency in our comparisons. For all other reported values, we directly source them from respective referenced papers.

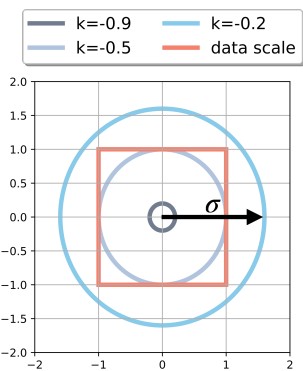

Figure 3: The standard deviaton $\sigma$ of the terminal marginal for uncontrolled dynamics. We empirically selected the hyperparameter $k = -0.2$. This choice induces a terminal marginal distribution with $\sigma$ that covers the data range with uncontrolled dynamics.

**Selection of $\boldsymbol{\Sigma}_0$**: The choice of initial covariance $\boldsymbol{\Sigma}_0$ directly influences the path measure of the trajectory. In our case, we set $\boldsymbol{\Sigma}_0 := \begin{bmatrix} 1 & k \\ k & 1 \end{bmatrix}$ with hyperparameter $k$. We observe that trajectories tend to exhibit pronounced curvature under specific conditions: when

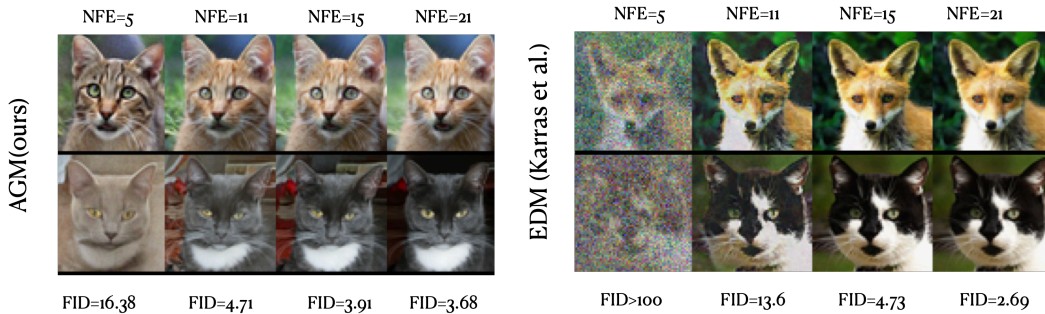

Figure 4: Comparison with EDM (Karras et al., 2022) on AFHQv2 dataset. AGM-ODE exhibits superior generative performance when NFE is exceedingly low, owing to its unique dynamics architecture that incorporates velocity when predicting the estimated data point.

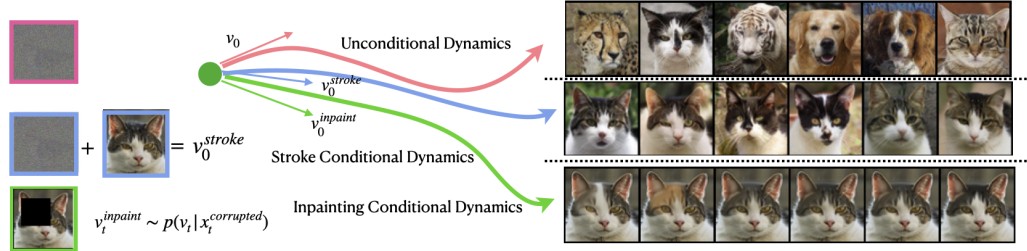

Figure 5: We showcase that AGM can generate conditional results from an unconditional model by injecting the conditional information into the velocity $\mathbf{v}_0$, thus leading to new initial velocity $\mathbf{v}_0^{cond}$.

the $k$ is positive, the absolute value of the position is large. This behavior is particularly noticeable when dealing with images, where the data scale ranges from -1 to 1. We aim for favorable uncontrolled dynamics, as this can potentially lead to better-controlled dynamics. Our strategy is to design $k$ in such a way that the marginal distribution of uncontrolled dynamics at $t_N = 1$ effectively covers the range of image data values meanwhile $k$ keeps negative. We can express the marginal of uncontrolled dynamics by leveraging the transition matrix $\Phi(1,0)$, which gives us $\mathbf{x}_1 := \mathbf{x}_0 + \mathbf{v}_0$. Figure 3 illustrates the standard deviation of $\mathbf{x}_1$ for various values of $k$. Based on our empirical observations, we choose $k = -0.2$ for all experiments, as it effectively covers the data range. The subsequent controlled dynamics (eq.7) will be constructed based on such desired uncontrolled dynamics as established.

**Stochastic Sampling:** In experiments, we emphasize the advantages of using the AGM-SDE compared with CLD. Firstly, we show that our model exhibits superior performance when NFE is significantly lower than that of CLD, particularly in toy dataset scenarios. For evaluation, we utilized the multi-modal Mixture of Gaussian and Multi-Swiss-Roll datasets. The results obtained from the toy dataset, as shown in Fig.8, demonstrate that AGM-SDE is capable of generating data that closely aligns with the ground truth, while requiring NFE that is around one order of magnitude lower than CLD. Furthermore, our findings reveal that AGM-SDE outperforms CLD in the context of CIFAR-10 image generation tasks, especially when faced with limited NFE, as illustrated in Table 2.

Table 2: FID↓ Comparison with CLD(Dockhorn et al., 2021) using same SSS Sampler on CIFAR-10.

| NFE↓ | CLD-SDE | AGM-SDE |
|------|---------|---------|
| 20   | >100    | **7.9** |
| 50   | 19.93   | **3.21** |
| 150  | 2.99    | **2.68** |
| 1000 | **2.44** | 2.46   |

**Deterministic Sampling:** We validate our algorithm on high-dimensional image generation with a deterministic sampler. We provide uncurated samples from CIFAR-10, AFHQv2 and ImageNet-64 with varying NFE in Appendix.H. Regarding the quantitative evaluation, Table.3 and Table.4 summarize the FID together with NFE used for sampling on CIFAR-10 and ImageNet-64. Notably, AGM-ODE achieves 2.46 FID score with 50 NFE on CIFAR-10, and 10.55 FID score with 20 NFE in unconditional ImageNet-64 which is comparable to the existing dynamical generative modeling.

Table 3: Unconditional CIFAR-10 generative performance

| | Model Name | NFE↓ | FID↓ |
|---|---|---|---|
| ODE | EDM (Karras et al., 2022) | 35 | 1.84 |
| | CLD+EI (Zhang et al., 2022) | 50 | 2.26 |
| | FM-OT (Lipman et al., 2022) | 142 | 6.35 |
| | **AGM-ODE(ours)** | 50 | 2.46 |
| SDE | VP (Song et al., 2020b) | 1000 | 2.66 |
| | VE (Song et al., 2020b) | 1000 | 2.43 |
| | CLD (Dockhorn et al., 2021) | 1000 | 2.44 |
| | **AGM-SDE(ours)** | 1000 | 2.46 |

Table 4: Unconditional ImageNet-64 generative performance

| Model | NFE↓ | FID↓ |
|---|---|---|
| FM-OT(Lipman et al., 2022) | 138 | 14.45 |
| MFM(Pooladian et al., 2023) | 132 | 11.82 |
| MFM(Pooladian et al., 2023) | 40 | 12.97 |
| **AGM-ODE(ours)** | 40 | 10.10 |
| **AGM-ODE(ours)** | 30 | 10.07 |
| **AGM-ODE(ours)** | 20 | 10.55 |

Table 5: Performance comparing with fast sampling algorithm using FID↓ metric on CIFAR-10

| Dynamics Order | Model Name | NFE↓ | 5 | 10 | 20 |
|---|---|---|---|---|---|
| 1st order dynamics | EDM (Karras et al., 2022) | | > 100 | 15.78 | **2.23** |
| | VP+EI (Zhang & Chen, 2022) | | 15.37 | **4.17** | 3.03 |
| | DDIM (Song et al., 2020a) | | 26.91 | 11.14 | 3.50 |
| | Analytic-DPM(Bao et al., 2022) | | 51.47 | 14.06 | 6.74 |
| 2nd order dynamics | CLD+EI (Zhang et al., 2022) | | N/A | 13.41 | 3.39 |
| | **AGM-ODE(ours)** | | **11.93** | 4.60 | 2.60 |

We underscore the effectiveness of sampling-hop, especially when faced with a constrained NFE budget, in comparison to baselines. We validate it on the CIFAR-10 and AFHQv2 dataset respectively. Fig.4 illustrates that AGM-ODE is able to generate plausible images even when NFE= 5 and outperforms EDM(Karras et al., 2022) when NFE is extremely small (NFE<15) visually and numerically on AFHQv2 dataset. We also compare with other fast sampling algorithms built upon DM in table.5 on CIFAR-10 dataset where AGM-ODE demonstrates competitive performance. Notably, AGM-ODE outperforms the baseline CLD with the same EI sampler by a large margin. We suspect that the improvement is based on the rectified trajectory which is more friendly for the ODE solver.

**Conditional Generation** We showcase the capability of AGM to generate conditional samples using an unconditional model (fig.5) by incorporating conditional information into the prior velocity variable $\mathbf{v}_0$. Instead of employing a randomly sampled $\mathbf{v}_0$, we use a linear combination of $\mathbf{v}_0$ and the desired velocity $\mathbf{v}_1 = (\mathbf{x}_1 - \mathbf{x}_{t_0})/(1 - t_0)$, where $\mathbf{x}_1$ is conditioned data. Thus, $t_0$, the initial velocity is defined as $\mathbf{v}_0^{cond} := (1 - \xi)\mathbf{v}_0 + \xi\mathbf{v}_1$, with $\xi$ serving as a mixing coefficient. Fig.5 shows that AGM can generate conditional data *without* augmentation and additional fine-tuning. Such property can be extended to the inpainting task as well and the detail can be found in appendix.F.

## 5 CONCLUSION AND LIMITATION

In this paper, we introduce a novel Acceleration Generative Modeling (AGM) framework rooted in SOC theory. Within this framework, we devise more favorable, straight trajectories for the momentum system. Leveraging the intrinsic characteristics of the momentum system, we capitalize on additional velocity to expedite the sampling process by using the sampling-hop technique, significantly reducing the time required to converge to accurate predictions of realistic data points. Our experimental results, conducted on both toy and image datasets in unconditional generative tasks, demonstrate promising outcomes for fast sampling.

However, it is essential to acknowledge that our approach's performance lags behind state-of-the-art methods in scenarios with sufficient NFE. This observation suggests avenues for enhancing AGM performance. Such improvements could be achieved by enhancing the training quality through the adoption of techniques proposed in Karras et al. (2022) including data augmentation, fine-tuned noise scheduling, and network preconditioning, among others.

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

## A  SUPPLEMENTARY SUMMARY

We state the assumptions in Appendix.B. We provide the technique details appearing in Section.3 at Appendix.D. The details of the experiments can be found in Appendix.E. The visualization of generated figures can be found in Appendix.H.

## B  ASSUMPTIONS

We will use the following assumptions to construct the proposed method. These assumptions are adopted from stochastic analysis for SGM (Song et al., 2021; Yong & Zhou, 1999; Anderson, 1982),

   (i) $p_0$ and $p_1$ with finite second-order moment.
   (ii) $g_t$ is continuous functions, and $|g(t)|^2 > 0$ is uniformly lower-bounded w.r.t. $t$.
   (iii) $\forall t \in [0, 1]$, we have $\nabla_{\mathbf{v}} \log p_t(\mathbf{m}_t, t)$ Lipschitz and at most linear growth w.r.t. $\mathbf{x}$ and $\mathbf{v}$.

Assumptions (i) (ii) are standard conditions in stochastic analysis to ensure the existence-uniqueness of the SDEs; hence also appear in SGM analysis (Song et al., 2021).

## C  STOCHASTIC OPTIMAL CONTROL (SOC) IN THE WILD

In this section, we are going to provide a gentle introduction of the Stochastic Optimal Control (SOC). Our work is majorly relying on the prior work Chen & Georgiou (2015) in which some technical details are missing. Here we first clarify some core derivations that may help the broader audience to understand Chen & Georgiou (2015) and our work.

### C.1  LINEAR QUADRATIC STOCHASTIC OPTIMAL CONTROL

SOC has wide applications in financial, robotics, and manufacturing. Here we will focus on Linear Quadratic SOC which usually refers to Linear Quadratic Regulator because the dynamic is linear and the objective function is quadratic (Bryson, 1975; Stengel, 1994). The problem states as:

$$\min_{\mathbf{u}_t} \int_0^1 \frac{1}{2} \|\mathbf{u}_t\|_2^2 \mathrm{d}t + \mathbf{x}_1^\mathsf{T} R \mathbf{x}_1$$

$$s.t \ \ \mathrm{d}\mathbf{x}_t = [A(t)\mathbf{x}_t + g_t \mathbf{u}_t]\mathrm{d}t + g_t \mathrm{d}w_t, \quad \mathbf{x}_0 = x_0. \tag{13}$$

In this formulation, $\mathbf{x}_t$ means the state, and $\mathbf{u}_t$ is the control variable. Conceptually, the SOC problem is aiming to design the controller $\mathbf{u}_t$ to drive the system from point $x_0$ to $x_1 \equiv 0$ with minimum effort. In the case of first order system, the control will be the optimal vector field $\mathbf{v}_t^*$ and for the second order system, the control is denoted as the optimal acceleration $\mathbf{a}_t^*$. The presence of stochasticity, introduced by the Wiener Process denoted as $\mathrm{d}w_t$, prevents the system from precisely converging to the Dirac mass $x_1$. In order to strike a balance between the objective of converging to $x_1$ and minimizing overall control effort $\int \|\mathbf{u}_t\|_2^2 \mathrm{d}t$, the terminal cost $\mathbf{x}_1^\mathsf{T} R \mathbf{x}_1$ has been imposed.

One special case is $R \to \infty$. Intuitively, it means the controlled dynamics should precisely converge to $x_1$. However, one can notice that the stochastic trajectory which connects $x_0$ and $x_1$ is not unique in this case. Based on this constraint (pinned down at $x_1$ and $x_0$ at two boundaries), the optimization problem of SOC finds the optimal solution with minimum effort $\mathbf{u}_t$ which can be understood as the regularization of the trajectories, hence, such stochastic trajectory is unique while the regularization of controller is still applied. One can also draw connection with such pinned-down SDE with well-known Doob-$h$ transform. For the people who are not familiar with these, here are some interesting paper (Heng et al., 2021; O'Connell, 2003).

The classical procedure to solve the SOC problem includes:

1. write down the Hamilton–Jacobi–Bellman equation (HJB PDE) which explicitly represents the propagation of value function over time.
2. Construct the Ricatti/Lyapunov Equation.
3. Solve Ricatti/Lyapunov Equation and obtain the optimal control.

## C.2  VALUE FUNCTION, HAMILTON-JACOBIAN (HAMILTON–JACOBI–BELLMAN EQUATION) AND RICATTI EQUATION

We adopt the classical notation in the SOC for the value function. Specifically, the underscript of the value function $V$ represents for the partial derivative of it. For example, $V_t$, $V_x$ and $V_{xx}$ represent for the first order derivative of $V$ w.r.t time $t$, state $\mathbf{x}$ and second order derivate of $V$ w.r.t $\mathbf{x}$. We first define the value function as:

$$V(\mathbf{x}_t, t) = \inf_{\mathbf{u}} \mathbb{E}\left[\int_t^1 \frac{1}{2}\|\mathbf{u}_t\|_2^2 d\tau + \mathbf{x}_1^\mathsf{T} R \mathbf{x}_1\right]$$

and the dynamics is,

$$d\mathbf{x}_t = (A\mathbf{x}_t + g_t \mathbf{u}_t)dt + g_t d\mathbf{w}_t$$

From Bellman's principle to the value function, one can get:

$$V(t, \mathbf{x}_t) = \inf_{\mathbf{u}} \mathbb{E}\left[V(t + dt, \mathbf{x}_{t+dt}) + \int_t^{t+dt} \frac{1}{2}\|\mathbf{u}_t\|_2^2 d\tau\right]$$

$$= \inf_{\mathbf{u}} \mathbb{E}\left[\frac{1}{2}\|\mathbf{u}_t\|_2^2 dt + V(t, \mathbf{x}_t) + V_t(t, \mathbf{x}_t)dt + V_x(t, \mathbf{x})d\mathbf{x} + \frac{1}{2}tr\left[V_{xx}gg^\mathsf{T}\right]dt\right]$$

$$= \text{Plug in the dynamics } d\mathbf{x}_t = \cdots$$

$$= \inf_{\mathbf{u}} \mathbb{E}\left[\frac{1}{2}\|\mathbf{u}_t\|_2^2 dt + V(t, \mathbf{x}_t) + V_t(t, \mathbf{x}_t)dt + V_x(t, \mathbf{x})^\mathsf{T}((A\mathbf{x}_t + g_t\mathbf{u}_t)dt + gd\mathbf{w}_t)\right.$$

$$\left. + \frac{1}{2}tr\left[V_{xx}gg^\mathsf{T}\right]dt\right]$$

$$= \inf_{\mathbf{u}}\left[\frac{1}{2}\|\mathbf{u}_t\|_2^2 dt + V(t, \mathbf{x}_t) + V_t(t, \mathbf{x}_t)dt + V_x(t, \mathbf{x})^\mathsf{T}(A\mathbf{x}_t + g_t\mathbf{u}_t)dt\right.$$

$$\left. + \frac{1}{2}tr\left[V_{xx}gg^\mathsf{T}\right]dt\right]$$

One obtain:

$$V_t + \inf_{\mathbf{u}}\left[\frac{1}{2}\|\mathbf{u}_t\|_2^2 + V_x^\mathsf{T}(A\mathbf{x}_t + g_t\mathbf{u}_t)\right] + \frac{1}{2}tr\left[V_{xx}gg^\mathsf{T}\right] = 0$$

The optimal control can be obtained by

$$\mathbf{u}_t^* = -g_t V_x$$

Plugging it back, one can obtain the HJB PDE:

$$V_t - \frac{1}{2}V_x gg^\mathsf{T} V_x + V_x^\mathsf{T} A\mathbf{x}_t + \frac{1}{2}tr\left[V_{xx}gg^\mathsf{T}\right] = 0$$

We assume that there exist certain matrix $Q$, s.t. $V(\mathbf{x}, t) \equiv \frac{1}{2}\mathbf{x}^\mathsf{T} Q\mathbf{x} + \Xi(t)$. By matching the different power term of HJB, one can write:

$$-\dot\Xi - \frac{1}{2}\mathbf{x}^\mathsf{T}\dot Q\mathbf{x} = -\frac{1}{2}\mathbf{x}^\mathsf{T} Q gg^\mathsf{T} Q\mathbf{x}^\mathsf{T} + \mathbf{x}^\mathsf{T} A^\mathsf{T} Q\mathbf{x} + \frac{1}{2}tr\left[Qgg^\mathsf{T}\right] \tag{14}$$

with boundary condition:

$$\Xi(1) = 0, \quad Q(1) = R \tag{15}$$

Due to the fact that $\mathbf{x}^\mathsf{T} A^\mathsf{T} Q\mathbf{x} = \mathbf{x}^\mathsf{T} QA\mathbf{x}$, one arrives Riccati Equation:

$$\boxed{-\dot Q = A^\mathsf{T} Q + QA - Qgg^\mathsf{T} Q} \tag{16}$$

Recall that the optimal solution is $\mathbf{u}_t^* = -g_t V_x$ and $V := \frac{1}{2}\mathbf{x}^\mathsf{T} Q\mathbf{x} + \Xi(t)$, the optimal control can be expressed in the way of the solution of Riccati equation: $\mathbf{u}_t^* = -g^\mathsf{T} Q(t)\mathbf{x}_t$.

### C.3 RICATTI EQUATION AND LYAPUNOV EQUATION

Here we provide the connection between Ricatti Equation and Lyapunov Equation in the current setup.

**Lemma 6.** *Define $P(t) := Q(t)^{-1}$ in which $Q(t)$ is the solution of Ricatti equation (eq.16), Then $P(t)$ solve the Lyapunov equation:*

$$\dot{P} = AP + PA^\mathsf{T} - gg^\mathsf{T} \tag{17}$$

For notation consistency, we name the elements in $P$ matrix as,

$$P = \begin{bmatrix} P_{00} & P_{01} \\ P_{10} & P_{11} \end{bmatrix}$$

*Proof.* By plugging in the Lyapunov equation $P(t) := Q(t)^{-1}$, one can get:

$$\dot{Q^{-1}} = AQ^{-1} + Q^{-1}A^\mathsf{T} - gg^\mathsf{T}$$
$$\Leftrightarrow -Q^{-1}\dot{Q}Q^{-1} = AQ^{-1} + Q^{-1}A^\mathsf{T} - gg^\mathsf{T}$$
$$\Leftrightarrow -\dot{Q} = QA + A^\mathsf{T}Q - Qgg^\mathsf{T}Q$$

$\square$

By Lemma.6, the optimal control can also be represented as the solution of the Lyapunov equation: $\mathbf{u}_t^* = -g^\mathsf{T}P(t)^{-1}\mathbf{x}_t$ which is indeed the optimal control term used in Chen & Georgiou (2015) after adopting their notation, and it is same as the optimal control term we used in the Lemma.12 without base dynamics compensation.

### C.4 SOC CONNECTION WITH SCHRÖDINGER BRIDGE

The optimal control solution is also the solution of Schrödinger Bridge when the terminal condition degenerate to the point mass (see example of Brownian Bridge in Appendix.D.1). It is also the solution of the Schrödinger Bridge when the optimal pairing is available see proposition.2 De Bortoli et al. (2023).

So in our case, we are **not** solving the momentum Schrödinger Bridge as shown in Chen et al. (2023) (also see. fig.6), even tough the problem formulation is similar. Specifically, AGM is a special case of momentum Schrödinger Bridge when the boundary conditions are degenerated to Dirac Distribtuions.

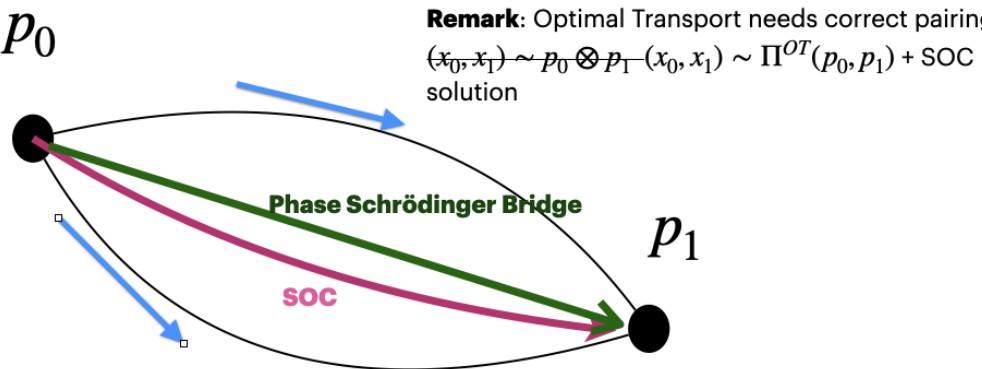

Figure 6: momentum Schrodinger Bridge versus AGM.

# D  TECHNIQUE DETAILS IN SECTION.3

## D.1  BROWNIAN BRIDGE AS THE SOLUTION OF STOCHASTIC OPTIMAL CONTROL

We adopt the presentation form Kappen (2008). We consider the control problem:

$$\min_{\mathbf{u}_t} \int_t^1 \frac{1}{2}\|\mathbf{u}_t\|_2^2 dt + \frac{\mathbf{r}}{2}\|\mathbf{x}_1 - x_1\|_2^2$$
$$\text{s.t.} \quad d\mathbf{x}_t = \mathbf{u}_t dt, \quad \mathbf{x}_0 = x_0$$

Where $\mathbf{r}$ is the terminal cost coefficient. According to Pontryagin Maximum Principle (PMP;Kirk (2004)) recipe, one can construct the Hamiltonian:

$$H(t, \mathbf{x}, \mathbf{u}, \gamma) = -\frac{1}{2}\|\mathbf{u}_t\|_2^2 + \gamma \mathbf{u}_t$$

By setting:

$$\frac{\partial H}{\partial \mathbf{u}_t} = 0,$$

the optimized Hamiltonian is:

$$H(t, \mathbf{x}, \mathbf{u}, \gamma)^* = \frac{1}{2}\gamma^2, \quad \text{where} \quad \mathbf{u}_t = \gamma$$

Then we solve the Hamiltonian equation of motion:

$$\frac{d\mathbf{x}_t}{dt} = \frac{\partial H^*}{\partial \gamma} = \gamma$$
$$\frac{d\gamma}{dt} = \frac{\partial H^*}{\partial \mathbf{x}} = 0$$
$$\text{where} \quad \mathbf{x}_0 = x_0 \quad \text{and} \quad \gamma_1 = -\mathbf{r} \cdot (\mathbf{x}_1 - x_1)$$

One can notice that the solution for $\gamma_t$ is the constant $\gamma_t = \gamma = -\mathbf{r} \cdot (\mathbf{x}_1 - x_1)$, hence the solution for $\mathbf{x}_t$ is $\mathbf{x}_t = \mathbf{x}_1 + \gamma t$.

$$\gamma = -\mathbf{r}(\mathbf{x}_1 - x_1) = -\mathbf{r}(\mathbf{x}_0 + (1-t)\gamma - x_1)$$
$$\rightarrow \quad \mathbf{u}_t^* := \gamma = \frac{\mathbf{r}(x_1 - \mathbf{x}_0)}{1 + \mathbf{r}(1-t)}$$

When $\mathbf{r} \rightarrow +\infty$, we arrive the optimal control as $\mathbf{u}_t^* = \frac{x_1 - \mathbf{x}_0}{1-t}$. Due to certainty equivalence, this is also the optimal control law for

$$d\mathbf{x}_t = \mathbf{u}_t dt + d\mathbf{w}_t$$

By plugging it back into the dynamics, we obtain the well-known Brownian Bridge:

$$d\mathbf{x}_t = \frac{x_1 - \mathbf{x}_t}{1-t}dt + d\mathbf{w}_t$$

**Remark 7.** *If there is not stochasticity $d\mathbf{w}_t$, one can get $\mathbf{u}_t := \frac{x_1 - \mathbf{x}_t}{1-t} = x_1 - \mathbf{x}_0$ which is the vector field constructed by Lipman et al. (2022) during **traning**.*

## D.2  PROOF OF PROPOSITION.3

**Proposition 8.** *The solution of the stochastic bridge problem of linear momentum system (Chen & Georgiou, 2015) is*

$$\mathbf{a}^*(\mathbf{m}_t, t) = g_t^2 P_{11}\left(\frac{\mathbf{x}_1 - \mathbf{x}_t}{1-t} - \mathbf{v}_t\right) \quad where: \quad P_{11} = \frac{-4}{g_t^2(t-1)}. \tag{18}$$

*Proof.* From Lemma.12, one can get the optimal control for this problem is

$$\mathbf{u}_t^* = -\mathbf{gg}^\mathsf{T}\mathbf{P}_t^{-1}\left(\mathbf{m}_t - \Phi(t,1)\mathbf{m}_1\right)$$

where state transition function $\Phi$ can be obtained from Lemma.11 and $\mathbf{P}_t$ is the solution of Lyapunov equation and $\mathbf{P}_t^{-1}$ can be found in Lemma.9.

Then we have:

$$
\begin{aligned}
\mathbf{u}_t^* &= -\mathbf{gg}^\mathsf{T}\mathbf{P}_t^{-1}\left(\mathbf{m}_t - \Phi(t,1)\mathbf{m}_1\right)\\
&= -\mathbf{gg}^\mathsf{T}\mathbf{P}_t^{-1}\mathbf{m}_t + \mathbf{gg}^\mathsf{T}\mathbf{P}_t^{-1}\Phi(t,1)\mathbf{m}_1\\
&= -\begin{bmatrix} 0 & 0 \\ 0 & g^2 \end{bmatrix}\mathbf{P}_t^{-1}\mathbf{m}_t + \mathbf{gg}^\mathsf{T}\mathbf{P}_t^{-1}\begin{bmatrix} 1 & t-1 \\ 0 & 1 \end{bmatrix}\mathbf{m}_1\\
&= -g_t^2\begin{bmatrix} 0 & 0 \\ P_{10} & P_{11} \end{bmatrix}\mathbf{m}_t + \begin{bmatrix} 0 & 0 \\ 0 & g_t^2 \end{bmatrix}\begin{bmatrix} P_{00} & P_{01} \\ P_{10} & P_{11} \end{bmatrix}\begin{bmatrix} 1 & t-1 \\ 0 & 1 \end{bmatrix}\mathbf{m}_1\\
&= -g_t^2\begin{bmatrix} 0 & 0 \\ P_{10} & P_{11} \end{bmatrix}\mathbf{m}_t + g_t^2\begin{bmatrix} 0 & 0 \\ P_{10} & P_{11} \end{bmatrix}\begin{bmatrix} 1 & t-1 \\ 0 & 1 \end{bmatrix}\mathbf{m}_1\\
&= -g_t^2\begin{bmatrix} 0 & 0 \\ P_{10} & P_{11} \end{bmatrix}\mathbf{m}_t + g_t^2\begin{bmatrix} 0 & 0 \\ P_{10} & P_{10}(t-1)+P_{11} \end{bmatrix}\mathbf{m}_1\\
&= \begin{bmatrix} 0 \\ g_t^2 P_{10}(\mathbf{x}_1 - \mathbf{x}_t) + g_t^2 P_{10}(t-1)\cdot\mathbf{v}_1 + g_t^2 P_{11}(\mathbf{v}_1 - \mathbf{v}_t) \end{bmatrix}
\end{aligned}
$$

Plug in $\mathbf{v}_1 := \dfrac{\mathbf{x}_1 - \mathbf{x}_t}{1-t}$

$$
= \begin{bmatrix} 0 \\ g_t^2 P_{11}\left(\frac{\mathbf{x}_1-\mathbf{x}_t}{1-t} - \mathbf{v}_t\right) \end{bmatrix}
$$

$\square$

**Lemma 9.** *The Lyapunov equation corresponding to the optimization problem showed in Lemma.12:*

$$
\mathbf{u}_t^* \in \operatorname*{arg\,min}_{\mathbf{u}_t\in\mathcal{U}}\mathbb{E}\left[\int_0^T \frac{1}{2}\|\mathbf{u}_t\|^2\right]\mathrm{d}t + \mathbf{x}_1^\mathsf{T}\mathbf{R}\mathbf{x}_1
$$

$$
s.t \quad \mathrm{d}\mathbf{m}_t = \underbrace{\begin{bmatrix} 0 & 1 \\ 0 & 0 \end{bmatrix}}_{A}\mathbf{m}_t\mathrm{d}t + \mathbf{u}_t\mathrm{d}t + \mathbf{g}\mathrm{d}\mathbf{w}_t
$$

$$
\mathbf{m}_0 = m_0, \quad \mathbf{m}_1 = m_1
$$

*is depited as*

$$
\dot{\mathbf{P}} = A\mathbf{P} + \mathbf{P}A^\mathsf{T} - \mathbf{gg}^T. \tag{19}
$$

*When* $\mathbf{g} = \begin{bmatrix} 0 \\ g \end{bmatrix}$, *the solution for Lyapunov equation above, with terminal condition*

$$
\mathbf{P}_1 = \mathbf{R}^{-1} = \lim_{\mathbf{r}\to\inf}\begin{bmatrix} \mathbf{r} & 0 \\ 0 & \mathbf{r} \end{bmatrix}^{-1} = \begin{bmatrix} 0 & 0 \\ 0 & 0 \end{bmatrix} \tag{20}
$$

*However, one does not need the force to converge exactly at* $\mathbf{v}_1$ *because we only care about the generate quality of* $\mathbf{x}_1$. *Here we give a general case in which the* $\mathbf{r}$ *keeps a small value* $\omega$ *for the velocity channel:*

$$
\mathbf{P}_1 = \mathbf{R}^{-1} = \begin{bmatrix} 0 & 0 \\ 0 & \omega \end{bmatrix} \tag{21}
$$

*Then the solution is given by*

$$\mathbf{P}_t = \begin{bmatrix} \omega(t-1)^2 - \frac{1}{3}g^2(t-1)^3 & \omega(t-1) - \frac{1}{2}g^2(t-1)^2 \\ \omega(t-1) - \frac{1}{2}g^2(t-1)^2 & g^2(1-t) + \omega \end{bmatrix}$$

*and the inverse of $\mathbf{P}_t$ is,*

$$\mathbf{P}_t^{-1} = \frac{1}{g^2(-4\omega + g^2(t-1))(t-1)} \begin{bmatrix} \frac{12(\omega - g^2(t-1))}{(t-1)^2} & \frac{6(-2\omega + g^2(t-1))}{t-1} \\ \frac{6(-2\omega + g^2(-1+t))}{t-1} & 12\omega - 4g^2(t-1) \end{bmatrix}$$

*Thus,*

$$P_{10} = \frac{-12\omega + 6g^2(t-1)}{g^2[-4\omega + g^2(t-1)](t-1)^2} = \frac{-12\omega}{g^2[-4\omega + g^2(t-1)](t-1)^2} + \frac{6}{[-4\omega + g^2(t-1)](t-1)}$$

$$P_{11} = \frac{12\omega - 4g^2(t-1)}{g^2[-4\omega + g^2(t-1)](t-1)} = \frac{12\omega}{g^2[-4\omega + g^2(t-1)](t-1)} + \frac{-4}{[-4\omega + g^2(t-1)]}$$

*Proof.* One can plug in the solution of $\mathbf{P}_t$ into the Lyapunov equation $\mathbf{P}_t$ and it validates $\mathbf{P}_t$ is indeed the solution.

**Remark 10.** *Here we provide a general form when the terminal condition of the Lyapunov function is not a zero matrix. It explicitly means that it allows that the velocity does not necessarily need to converge to the exact predefined $\mathbf{v}_1$. It will have the same results as shown in the paper by setting $\omega = 0$.*

$\square$

**Lemma 11.** *The state transition function $\Phi(t, s)$ of following dynamics,*

$$\mathbf{dm}_t = \begin{bmatrix} 0 & 1 \\ 0 & 0 \end{bmatrix} \mathbf{m}_t \mathbf{d}t$$

*is,*

$$\Phi(t, s) = \begin{bmatrix} 1 & t - s \\ 0 & 1 \end{bmatrix}$$

*Proof.* One can easily verify that such $\Phi$ satisfies $\partial\Phi/\partial t = \begin{bmatrix} 0 & 1 \\ 0 & 0 \end{bmatrix} \Phi$. $\square$

**Lemma 12** (Chen & Georgiou (2015))**.** *When $R \to \infty$, The optimal control $\mathbf{u}_t^*$ of following problem,*

$$\mathbf{u}_t^* = \begin{bmatrix} \mathbf{0} \\ \mathbf{a}_t \end{bmatrix} \in \arg\min_{\mathbf{u}_t \in \mathcal{U}} \int_0^T \frac{1}{2} \|\mathbf{u}_t\|^2 \mathbf{d}t + \mathbf{x}_1^\mathsf{T} R \mathbf{x}_1$$

$$s.t \quad \mathbf{dm}_t = \begin{bmatrix} 0 & 1 \\ 0 & 0 \end{bmatrix} \mathbf{m}_t \mathbf{d}t + \mathbf{u}_t \mathbf{d}t + \mathbf{g}_t \mathbf{dw}_t$$

$$\mathbf{m}_0 = m_0$$

*is given by*

$$\mathbf{u}_t^* = -\mathbf{gg}^\mathsf{T} \mathbf{P}_t^{-1} (\mathbf{m}_t - \Phi(t, 1)\mathbf{m}_1)$$

*Where $\mathbf{P}_t$ follows Lyapunov equation (eq.19) with boundary condition $\mathbf{P}_1 = \mathbf{0}$. and function $\Phi(t, s)$ is the transition matrix from time-step $s$ to time-step $t$ given uncontrolled dynamics.*

*And it is indeed the stochastic bridge of following system:*

$$\mathbf{dm}_t = \begin{bmatrix} 0 & 1 \\ 0 & 0 \end{bmatrix} \mathbf{m}_t \mathbf{d}t + \mathbf{u}_t \mathbf{d}t + g\mathbf{dw}_t \tag{22}$$

$$\mathbf{m}_0 = m_0, \quad \mathbf{m}_1 = m_1 \tag{23}$$

*Proof.* See page 8 in Chen & Georgiou (2015). $\square$

### D.3 MEAN AND COVARIANCE OF SDE

By plugging the optimal control into the system, one can obtain the system as:

$$
\begin{aligned}
\mathbf{dm}_t &= \begin{bmatrix} \mathbf{v}_t \\ \mathbf{F}_t \end{bmatrix} dt + \mathbf{g}_t \mathbf{dw}_t \\
&= \begin{bmatrix} \mathbf{v}_t \\ g_t^2 P_{11} \left( \frac{\mathbf{x}_1 - \mathbf{x}_t}{1-t} - \mathbf{v}_t \right) \end{bmatrix} dt + \mathbf{g}_t \mathbf{dw}_t \\
&= \underbrace{\begin{bmatrix} \mathbf{0} & \mathbf{1} \\ -\frac{g_t^2 P_{11}}{1-t} & -g_t^2 P_{11} \end{bmatrix}}_{\tilde{\mathbf{F}}_t} \begin{bmatrix} \mathbf{x}_t \\ \mathbf{v}_t \end{bmatrix} dt + \underbrace{\begin{bmatrix} \mathbf{0} \\ \frac{g_t^2 P_{11}}{1-t} \mathbf{x}_1 \end{bmatrix}}_{\tilde{\mathbf{D}}_t} dt + \mathbf{g}_t \mathbf{dw}_t
\end{aligned}
$$

We follow the recipe of Särkkä & Solin (2019). The mean $\boldsymbol{\mu}_t$ and variance $\boldsymbol{\Sigma}_t$ of the matrix of random variable $\mathbf{m}_t$ obey the following respective ordinary differential equations (ODEs):

$$
d\boldsymbol{\mu}_t = \tilde{\mathbf{F}}_t \boldsymbol{\mu}_t dt + \tilde{\mathbf{D}}_t dt
$$

$$
d\boldsymbol{\Sigma}_t = \tilde{\mathbf{F}}_t \boldsymbol{\Sigma}_t dt + \left[ \tilde{\mathbf{F}}_t \boldsymbol{\Sigma}_t \right]^{\mathsf{T}} dt + \mathbf{gg}^{\mathsf{T}} dt
$$

One can solve it by numerically simulating two ODEs whose dimension is just two. Or one can use software such as Inc. (2022) to get analytic solutions. If you opt to the later approach, you can get:

$$
\mu_t^x = \frac{1}{3} \mathbf{x}_1 t^2 (t^2 - 4t + 6)
$$

$$
\mu_t^v = \frac{4t\mathbf{x}_1}{3} (t^2 - 3t + 3)
$$

$$
\Sigma_t^{xx} = -\frac{1}{9} \left\{ (-1+t)^2 \left[ -9 + 2(-1+k)t \left( 3 + (-3+t)t \right) \left( 3 + t \left[ 3 + (-3+t)t \right] \right) \right] \right\}
$$

$$
\Sigma_t^{xv} = \frac{1}{9} \left\{ (-1+t) \left[ t \left( 3 + (-3+t)t \right) \left( 9 + 8t \left( 3 + (-3+t)t \right) \right) + k \left( 9 - t \left( 3 + (-3+t)t \right) \left( 9 + 8t \left( 3 + (-3+t)t \right) \right) \right) \right] \right\}
$$

$$
\Sigma_t^{vv} = 1 - \frac{8}{9} (-1+k)t \left[ 3 + (-3+t)t \right] \left\{ -3 + 4t \left( 3 + (-3+t)t \right) \right\}
$$

**Remark 13.** *The expressions above are too complicated. Hence, we provide the python functional bracket in Appendix.E.1 with general initial covariance and diffusion coefficient for easy copy-paste. Equations above are ones we used through this paper and feel free to play around with other hyper-parameters.*

### D.4 DERIVATION FROM SDE TO ODE FOR PHASE DYNAMICS

One can represent the dynamics in the form of,

$$
\begin{bmatrix} \mathbf{dx}_t \\ \mathbf{dv}_t \end{bmatrix} = \begin{bmatrix} \mathbf{v}_t \\ \mathbf{F}_t \end{bmatrix} dt + \begin{bmatrix} \mathbf{0} & \mathbf{0} \\ \mathbf{0} & g_t \end{bmatrix} \mathbf{dw}_t \quad \text{s.t} \quad \mathbf{m}_0 := \begin{bmatrix} \mathbf{x}_0 \\ \mathbf{v}_0 \end{bmatrix} \sim \mathcal{N}(\boldsymbol{\mu}_0, \boldsymbol{\Sigma}_0) \tag{24}
$$

as

$$
\mathbf{dm}_t = f(\mathbf{m}_t) dt + \mathbf{g}_t \mathbf{dw}_t
$$

And its corresponding Fokker-Planck Partial Differential Equation Øksendal (2003) reads,

$$
\frac{\partial p_t}{\partial t} = -\sum_d \frac{\partial}{\partial \mathbf{m}_i} [f_i(\mathbf{m}, t) p_t(\mathbf{m}_t)] + \frac{1}{2} \sum_d \frac{\partial^2}{\partial \mathbf{m}_i \mathbf{m}_j} \left[ \sum_d \mathbf{g}_t \mathbf{g}_t^{\mathsf{T}} p_t(\mathbf{m}_t) \right] \tag{25}
$$

According to eq.(37) in Song et al. (2020b), One can rewrite such PDE,

$$\frac{\partial p_t}{\partial t} = -\sum_d \frac{\partial}{\partial \mathbf{m}_i} \left\{ f_i(\mathbf{m}_t, t) p_t(\mathbf{m}_t) - \frac{1}{2} \left[ p(\mathbf{m}_t) \nabla_{\mathbf{m}} \cdot (\mathbf{g}_t \mathbf{g}_t^\mathsf{T}) + p(\mathbf{m}_t) \mathbf{g}_t \mathbf{g}_t^\mathsf{T} \nabla_{\mathbf{m}} \log p(\mathbf{m}_t) \right] \right\}$$
(26)

due to the fact $\mathbf{g}_t \equiv \begin{bmatrix} \mathbf{0} & \mathbf{0} \\ \mathbf{0} & g_t \end{bmatrix}$
(27)

$$= -\sum_d \frac{\partial}{\partial \mathbf{m}_i} \left\{ f_i(\mathbf{m}_t, t) p_t(\mathbf{m}_t) - \frac{1}{2} p(\mathbf{m}_t) \left[ g_t^2 \nabla_{\mathbf{v}} \log p(\mathbf{m}_t) \right] \right\}$$
(28)

Then one can get the equivalent ODE:

$$d\mathbf{m}_t = \left[ f(\mathbf{m}_t, t) - \frac{1}{2} g_t^2 \nabla_{\mathbf{v}} \log p(\mathbf{m}, t) \right] dt$$
(29)

## D.5   DECOMPOSITION OF COVARIANCE MATRIX AND REPRESENTATION OF SCORE

Here we follow the procedure in Dockhorn et al. (2021). Given the covariance matrix $\mathbf{\Sigma}_t$, the decomposition of the positive definite symmetric matrix is,

$$\mathbf{\Sigma}_t = \mathbf{L}_t^\mathsf{T} \mathbf{L}_t$$
(30)

Where,

$$\mathbf{L}_t = \begin{bmatrix} L_t^{xx} & L_t^{xv} \\ L_t^{xv} & L_t^{vv} \end{bmatrix} = \begin{bmatrix} \sqrt{\Sigma_t^{xx}} & 0 \\ \frac{\Sigma_t^{xv}}{\sqrt{\Sigma_t^{xx}}} & \sqrt{\frac{\Sigma_t^{xx}\Sigma_t^{vv} - \Sigma_t^{vv}}{\Sigma_t^{xx}}} \end{bmatrix}$$
(31)

We borrow results from Dockhorn et al. (2021), the score function reads,

$$\nabla_{\mathbf{m}} \log p(\mathbf{m}_t | \mathbf{m}_1) = -\nabla_{\mathbf{m}_t} \frac{1}{2} (\mathbf{m}_t - \boldsymbol{\mu}_t) \mathbf{\Sigma}_t^{-1} (\mathbf{m}_t - \boldsymbol{\mu}_t)$$
$$= -\mathbf{\Sigma}_t^{-1} (\mathbf{m}_t - \boldsymbol{\mu}_t)$$
$$\text{Cholesky decomposition of } \mathbf{\Sigma}_t$$
$$= -\mathbf{L}^{-T} \mathbf{L}^{-1} (\mathbf{m}_t - \boldsymbol{\mu}_t)$$
$$= -\mathbf{L}^{-T} \epsilon$$

The form of $\mathbf{L}$ reads,

$$\mathbf{L}_t = \begin{bmatrix} \sqrt{\Sigma_t^{xx}} & 0 \\ \frac{\Sigma_t^{xv}}{\sqrt{\Sigma_t^{xx}}} & \sqrt{\frac{\Sigma_t^{xx}\Sigma_t^{vv} - (\Sigma_t^{xv})^2}{\Sigma_t^{xx}}} \end{bmatrix}$$

and the transpose inverse of $\mathbf{L}$ reads,

$$\mathbf{L}_t^{-T} = \begin{bmatrix} \frac{1}{\sqrt{(\Sigma_t^{xx} + \epsilon_{xx})}} & \frac{-\Sigma_t^{xv}}{\sqrt{(\Sigma_t^{xx})}\sqrt{(\Sigma_t^{xx})(\Sigma_t^{vv}+) - (\Sigma_t^{xv})^2}} \\ 0 & \frac{\sqrt{\Sigma_t^{xx}}}{\sqrt{(\Sigma_t^{xx})(\Sigma_t^{vv}) - (\Sigma_t^{xv})^2}} \end{bmatrix}$$

Hence, the score function reads,

$$\nabla_{\mathbf{v}} \log p(\mathbf{m}_t | \mathbf{m}_1) = -\underbrace{\frac{\sqrt{\Sigma_t^{xx}}}{\sqrt{(\Sigma_t^{xx} + \epsilon_{xx})(\Sigma_t^{vv} + \epsilon_{vv}) - (\Sigma_t^{xv})^2}}}_{\ell_t} \epsilon_1$$

## D.6 REPRESENTATION OF ACCELERATION $\mathbf{a}_t$

As been shown in Proposition.3, the optimal control can be represented as,

$$\mathbf{a}_t^* = g_t^2 P_{11} \left( \frac{\mathbf{x}_1 - \mathbf{x}_t}{1-t} - \mathbf{v}_t \right)$$

$$= g_t^2 P_{11} \frac{\mathbf{x}_1}{1-t} - g_t^2 P_{11} \left( \frac{\mathbf{x}_t}{1-t} + \mathbf{v}_t \right)$$

$$= g_t^2 P_{11} \frac{\mathbf{x}_1}{1-t} - g_t^2 P_{11} \left( \frac{\mu_t^x + L_t^{xx} \boldsymbol{\epsilon}_0}{1-t} + (\mu_t^v + L_t^{xv} \boldsymbol{\epsilon}_0 + L_t^{vv} \boldsymbol{\epsilon}_1) \right)$$

$$= g_t^2 P_{11} \left[ \left( \frac{\mathbf{x}_1 - \mu_t^x}{1-t} - \mu_t^v \right) - \left( \frac{L_t^{xx}}{1-t} \boldsymbol{\epsilon}_0 + L_t^{xv} \boldsymbol{\epsilon}_0 + L_t^{vv} \boldsymbol{\epsilon}_1 \right) \right]$$

solving eq.D.3 we can get : $\mu_t^x = \frac{1}{3} \mathbf{x}_1 t^2 (t^2 - 4t + 6), \mu_t^v = \frac{4t\mathbf{x}_1}{3}(t^2 - 3t + 3)$

Plug in$\mathbf{x}_t, \mathbf{v}_t$

$$= g_t^2 P_{11} \left[ \left( \frac{\mathbf{x}_1 - \frac{1}{3}\mathbf{x}_1 t^2 \left( 6 - 4t + t^2 \right)}{1-t} - \frac{4t\mathbf{x}_1}{3}(t^2 - 3t + 3) \right) - \left( \frac{L_t^{xx}}{1-t} \boldsymbol{\epsilon}_0 + L_t^{xv} \boldsymbol{\epsilon}_0 + L_t^{vv} \boldsymbol{\epsilon}_1 \right) \right]$$

$$= g_t^2 P_{11} \left[ \left( \frac{(-t^4 + 4t^3 - 6t^2 + 3)}{3(1-t)} - \frac{4t}{3}(t^2 - 3t + 3) \right) \mathbf{x}_1 - \left( \frac{L_t^{xx}}{1-t} \boldsymbol{\epsilon}_0 + L_t^{xv} \boldsymbol{\epsilon}_0 + L_t^{vv} \boldsymbol{\epsilon}_1 \right) \right]$$

$$= g_t^2 P_{11} \left[ \left( \frac{-(t-1)(t^3 - 3t^2 + 3t + 3)}{3(1-t)} - \frac{4t}{3}(t^2 - 3t + 3) \right) \mathbf{x}_1 - \left( \frac{L_t^{xx}}{1-t} \boldsymbol{\epsilon}_0 + L_t^{xv} \boldsymbol{\epsilon}_0 + L_t^{vv} \boldsymbol{\epsilon}_1 \right) \right]$$

$$= g_t^2 P_{11} \left[ \left( \frac{(t^3 - 3t^2 + 3t + 3)}{3} - \frac{1}{3}(4t^3 - 12t^2 + 12t) \right) \mathbf{x}_1 - \left( \frac{L_t^{xx}}{1-t} \boldsymbol{\epsilon}_0 + L_t^{xv} \boldsymbol{\epsilon}_0 + L_t^{vv} \boldsymbol{\epsilon}_1 \right) \right]$$

$$= g_t^2 P_{11} \left[ (1-t)^3 \mathbf{x}_1 - \left( \frac{L_t^{xx}}{1-t} \boldsymbol{\epsilon}_0 + L_t^{xv} \boldsymbol{\epsilon}_0 + L_t^{vv} \boldsymbol{\epsilon}_1 \right) \right]$$

$$= 4(1-t)^2 \mathbf{x}_1 + g_t^2 P_{11} \left( \frac{L_t^{xx}}{1-t} \boldsymbol{\epsilon}_0 + L_t^{xv} \boldsymbol{\epsilon}_0 + L_t^{vv} \boldsymbol{\epsilon}_1 \right)$$

## D.7 LOSS REWEIGHT

In practice, we use the following loss function

$$\mathcal{L} = \min_\theta \mathbb{E}_{t \in [0,1]} \mathbb{E}_{\mathbf{x}_1 \sim p_{\text{data}}} \mathbb{E}_{\mathbf{m}_t \sim p_t(\mathbf{m}_t | \mathbf{x}_1)} \lambda(t) \left[ \| \mathbf{F}_t^\theta(\mathbf{m}_t, t; \theta) - \mathbf{F}_t(\mathbf{m}_t, t) \|_2^2 \right] \qquad (32)$$

$$\propto \min_\theta \mathbb{E}_{t \in [0,1]} \mathbb{E}_{\mathbf{x}_1 \sim p_{\text{data}}} \mathbb{E}_{\mathbf{m}_t \sim p_t(\mathbf{m}_t | \mathbf{x}_1)} \frac{1}{1-t} \left[ \| \mathbf{F}_t^\theta(\mathbf{m}_t, t; \theta) - \mathbf{F}_t(\mathbf{m}_t, t) / \mathbf{z}_t \|_2^2 \right] \qquad (33)$$

We admit that this might not be an optimal selection. The motivation behind this is simply increasing the weight of training when $t \to 1$ and normalize the label with normalizer $\mathbf{z}_t$.

## D.8 NORMALIZER OF AGM-SDE AND AGM-ODE

Since the optimal control term can be represented as,

$$\mathbf{a}^*(\mathbf{m}_t, t) = 4\mathbf{x}_1(1-t)^2 - g_t^2 P_{11} \left[ \left( \frac{L_t^{xx}}{1-t} + L_t^{xv} \right) \boldsymbol{\epsilon}_0 + L_t^{vv} \boldsymbol{\epsilon}_1 \right].$$

Then we introduce the normalizer as

$$\mathbf{z}_{SDE} = \sqrt{(4(1-t)^2 \cdot \sigma_{data})^2 + g_t^2 P_{11} \left[ \left( \frac{L_t^{xx}}{1-t} + L_t^{xv} \right)^2 + (L_t^{vv})^2 \right]}$$

$$\mathbf{z}_{ODE} = \sqrt{(4(1-t)^2 \cdot \sigma_{data})^2 + g_t^2 P_{11} + g_t^2 P_{11} \left( \frac{L_t^{xx}}{1-t} + L_t^{xv} \right)^2 + \left[ \left( g_t^2 P_{11} L_t^{vv} - \frac{1}{2} g_t^2 \ell_t \right)^2 \right]}$$

Where $\ell := \sqrt{\frac{\Sigma_t^{xx}}{\Sigma_t^{xx}\Sigma_t^{vv}-(\Sigma_t^{xv})^2}}$

## D.9 EXPONENTIAL INTEGRATOR DERIVATION

As suggested by Zhang & Chen (2022), one can write the discretized dynamics as,

$$\begin{bmatrix}\mathbf{x}_{t_{i+1}}\\\mathbf{v}_{t_{i+1}}\end{bmatrix} = \Phi(t_{i+1},t_i)\begin{bmatrix}\mathbf{x}_t\\\mathbf{v}_t\end{bmatrix} + \sum_{j=0}^r C_{i,j}\begin{bmatrix}\mathbf{0}\\\mathbf{s}_\theta(\mathbf{m}_{t_{i-j}},t_{i-j})\end{bmatrix}$$

$$\text{Where } C_{i,j} = \int_t^{t+\delta_t}\Phi(t+\delta_t,\tau)\begin{bmatrix}\mathbf{0}&\mathbf{0}\\\mathbf{0}&\mathbf{z}_\tau\end{bmatrix}\prod_{k\neq j}\begin{bmatrix}\tau-t_{i-k}\\t_{i-j}-t_{i-k}\end{bmatrix}\mathrm{d}\tau, \quad \Phi(t,s) = \begin{bmatrix}1&t-s\\0&1\end{bmatrix}$$

(34)

After plugging in the transition kernel $\Phi(t,s)$, one can easily obtain the results shown in (11).

**Remark 14.** *In light of the momentum system, there are numerous methods for achieving high accuracy in its resolution. However, the practical performance in generative modeling remains untested. We recommend that readers consult the classical numerical physics text book or recent momentum dynamics solver (Pandey et al., 2023; Dockhorn et al., 2021).*

## D.10 PROOF OF PROPOSITION.5

The estimated data point $\mathbf{x}_1$ can be represented as

$$\tilde{\mathbf{x}}_1^{SDE} = \frac{(1-t)(\mathbf{F}_t^\theta+\mathbf{v}_t)}{g_t^2 P_{11}} + \mathbf{x}_t, \quad \text{or} \quad \tilde{\mathbf{x}}_1^{ODE} = \frac{\mathbf{F}_t^\theta + g_t^2 P_{11}(\alpha_t\mathbf{x}_t+\beta_t\mathbf{v}_t)}{4(t-1)^2 + g_t^2 P_{11}(\alpha_t\mu_t^x+\beta_t\mu_t^v)}$$

(35)

for SDE and probablistic ODE dynamics respectively, and $\beta_t = L_t^{vv} + \frac{1}{2P_{11}}, \alpha_t = \frac{(\frac{L_t^{xx}}{1-t}+L_t^{xv})-\beta_t L_t^{xv}}{L_t^{xx}}$.

*Proof.* It is easy to derive the representation of $\mathbf{x}_1$ of the SDE due to the fact that the network is essentially estimating:

$$\mathbf{F}_t^\theta \approx g_t^2 P_{11}\left(\frac{\mathbf{x}_1-\mathbf{x}_t}{1-t}-\mathbf{v}_t\right)$$

$$\Leftrightarrow \mathbf{x}_1 \approx \frac{(1-t)(\mathbf{F}_t^\theta+\mathbf{v}_t)}{g_t^2 P_{11}} + \mathbf{x}_t$$

It will become slightly more complicated for probabilistic ODE cases. We notice that

$$\mathbf{m}_t = \boldsymbol{\mu}_t + \mathbf{L}\boldsymbol{\epsilon}$$

$$\Leftrightarrow \quad \mathbf{x}_t = \mu_t^x + L_t^{xx}\boldsymbol{\epsilon}_1, \quad \mathbf{v}_t = \mu_t^v + L_t^{xv}\boldsymbol{\epsilon}_0 + L_t^{vv}\boldsymbol{\epsilon}_1$$

In probabilistic ODE case, the force term can be represented as,

$$\mathbf{F}(\mathbf{m}_t,t) = 4\mathbf{x}_1(1-t)^2 - g_t^2 P_{11}\left[\left(\frac{L_t^{xx}}{1-t}+L_t^{xv}\right)\boldsymbol{\epsilon}_0 + L_t^{vv}\boldsymbol{\epsilon}_1\right] - \frac{1}{2}g_t^2\ell\boldsymbol{\epsilon}_1$$

In order to use linear combination of $\mathbf{x}_t$ and $\mathbf{v}_t$ to represent $\mathbf{F}$ one needs to match the stochastic term in $\mathbf{F}_t$ by using

$$\alpha_t L_t^{xx} + \beta_t L_t^{xv} = \underbrace{\frac{L_t^{xx}}{1-t}+L_t^{xv}}_{\hat{\zeta}_t},$$

$$\beta_t L_t^{vv} = \underbrace{L_t^{vv}+\frac{1}{2P_{11}}}_{\zeta_t}.$$

The solution can be obtained by:

$$\beta_t = \frac{\zeta_t}{L_t^{vv}}$$

$$\alpha_t = \frac{\hat{\zeta}_t - \beta_t L_t^{xv}}{L_t^{xx}}$$

By substitute it back to $\mathbf{F}_t$, one can get:

$$
\begin{aligned}
\mathbf{F}(\mathbf{m}_t, t) &= 4\mathbf{x}_1(1-t)^2 - g_t^2 P_{11}\left[\alpha_t(\mathbf{x}_t - \mu_t^x) + \beta_t(\mathbf{v}_t - \mu_t^v)\right] \\
&= \left[4(1-t)^2 + g_t^2 P_{11}(\alpha_t\mu_t^x + \beta_t\mu_t^v)\right]\mathbf{x}_1 - g_t^2 P_{11}\left[\alpha_t\mathbf{x}_t + \beta_t\mathbf{v}_t\right] \\
\Leftrightarrow \mathbf{x}_1 &= \frac{\mathbf{F}_t^\theta + g_t^2 P_{11}(\alpha_t\mathbf{x}_t + \beta_t\mathbf{v}_t)}{4(t-1)^2 + g_t^2 P_{11}(\alpha_t\mu_t^x + \beta_t\mu_t^v)}
\end{aligned}
$$

$\square$

# E  EXPERIMENTAL DETAILS

**Training:** We stick with hyperparameters introduced in the section.4. We use AdamW(Loshchilov & Hutter, 2017) as our optimizer and Exponential Moving Averaging with the exponential decay rate of 0.9999. We use $8 \times$ Nvidia A100 GPU for all experiments. For further, training setup, please refer to Table.6.

Table 6: Additional experimental details

| dataset | Training Iter | Learning rate | Batch Size | network architecture |
|---------|---------------|---------------|------------|---------------------|
| toy | 0.05M | 1e-3 | 1024 | ResNet(Dockhorn et al., 2021) |
| CIFAR-10 | 0.5M | 1e-3 | 512 | NCSN++(Karras et al., 2022) |
| AFHQv2 | 0.5M | 1e-3 | 512 | NCSN++(Karras et al., 2022) |
| ImageNet-64 | 1.6M | 2e-4 | 512 | ADM(Dhariwal & Nichol, 2021) |

**Sampling:** For Exponential Integrator, we choose the multistep order $w = 2$ consistently for all experiments. Different from previous work (Dockhorn et al., 2021; Karras et al., 2022; Zhang et al., 2023), we use quadratic timesteps scheme with $\kappa = 2$:

$$
t_i = \left(\frac{N-i}{N}t_0^{\frac{1}{\kappa}} + \frac{i}{N}t_N^{\frac{1}{\kappa}}\right)^\kappa
$$

Which is opposite to the classical DM. Namely, the time discretization will get larger when the dynamics is propagated close to data. For numerical stability, we use $t_0 = 1E-5$ for all experiments. For $NFE = 5$, we use $t_N = 0.5$ and $NFE = 10$, $T_N = 0.7$. For the rest of the sampling, we use $t_N = 0.999$.

Due to the fact that EDM(Karras et al., 2022) is using second-order ODE solver, in practice, we allow it to have an extra one NFE as reported for all the tables.

## E.1  CODE EXAMPLE FOR COVARIANCE

We will abuse the notation in this coding section. Here we provide the example code for compute the covariance matrix. Here we consider the general case where $\boldsymbol{\Sigma}_0 := \begin{bmatrix} m & -k\sqrt{mn} \\ -k\sqrt{mn} & n \end{bmatrix}$ and the diffusion coefficient is $g(t) := p(tt - t)$ where $p$ is the scaling coefficient and $tt$ is the damping coefficient.

```
def Sigmaxx(t,p,tt,m,n):
    return  \
    (t - 1)**2*(30*m*(t**3 - 3*t**2 + 3*t + 3)**2\
    - 60*p**2*(t - 1)**3*torch.log(1 - t) \
    - t*(60*k*np.sqrt(m*n)*(t**5 - 6*t**4 + 15*t**3 - 15*t**2 + 9)\
    - 30*n*t*(t**2 - 3*t + 3)**2 + p**2*(t**5*(6*tt**2 + 3*tt + 1) \
    - 6*t**4*(6*tt**2 + 3*tt + 1)\
    + 15*t**3*(6*tt**2 + 3*tt + 1)\
    - 10*t**2*(9*tt**2 + 11) + 150*t - 60)))/270
```

```
def Sigmaxv(t,p,tt,m,n):
    return  \
    (1/270 - t/270)*(30*k*np.sqrt(m*n)*(8*t**6 - 48*t**5\
    + 120*t**4 - 135*t**3 + 45*t**2 + 27*t - 9) +\
    150*p**2*(t - 1)**3*torch.log(1 - t)\
    + t*(-120*m*(t**5 - 6*t**4 + 15*t**3 - 15*t**2 + 9)\
    - 30*n*(4*t**5 - 24*t**4 + 60*t**3 - 75*t**2 + 45*t - 9)\
    + p**2*(4*t**5*(6*tt**2 + 3*tt + 1) - 24*t**4*(6*tt**2 + 3*tt + 1)\
    + 60*t**3*(6*tt**2 + 3*tt + 1) - 5*t**2*(81*tt**2 + 18*tt + 55)\
    + 15*t*(9*tt**2 + 25) - 150)))

def Sigmavv(t,p,tt,m,n):
    return  \
    n*(-4*t**3 + 12*t**2 - 12*t + 3)**2/9\
    - 8*p**2*(t - 1)**3*torch.log(1 - t)/9\
    + t*(-120*k*np.sqrt(m*n)*(4*t**5 - 24*t**4 + 60*t**3\
    - 75*t**2 + 45*t - 9) + 240*m*t*(t**2 - 3*t + 3)**2 \
    + p**2*(-8*t**5*(6*tt**2 + 3*tt + 1) + 48*t**4*(6*tt**2 + 3*tt + 1)\
    - 120*t**3*(6*tt**2 + 3*tt + 1) + 5*t**2*(180*tt**2 + 72*tt + 53)\
    - 15*t*(36*tt**2 + 9*tt + 20) + 135*tt**2 + 120))/135
```

# F CONDITIONAL GENERATION DETAILS

Here we provide the detail of conditional generation details.

## F.1 STORKE BASED GENERATION

For stroke based generation, we provide two types of conditional generation.

**initial Velocity (IV):**Please refer to section.4.
**Dynamics Velocity (dyn-V):**Since the mean and variance of velocity and position are available, one can specify the velocity which is valid. In this case, we can set the velocity as

$$v_t = \mu_t^{v_t|x_t} + \Sigma_t^{v_t|x_t}\epsilon \tag{36}$$

In which,

$$\mu_t^{v_t|x_t} = \mu_t^v + \frac{\Sigma_t^{xv}}{\Sigma_t^{xx}}(\mathbf{x}_t - \mu_t^x) \tag{37}$$

$$\Sigma_t^{v_t|x_t} = \Sigma_t^{vv} - \frac{\Sigma_t^{xv2}}{\Sigma_t^{xx}} \tag{38}$$

when $t \leq c$. The $c$ is the guidance length. We typically set it to be $c = 0.25$.

## F.2 INPAINTING

In the inpainting case, we apply the similar strategy as **dyn-V**. Specifically, in this case, the $\tilde{\mathbf{x}}_1$ will be represented as:

$$\hat{\mathbf{x}}_1 := \text{MASK} \odot \mu_t^x + (1 - \text{MASK}) \odot \tilde{\mathbf{x}}_1 \tag{39}$$

where MASK represents for the mask matrix which zero-out the pixel of the original image. Such $\hat{\mathbf{x}}_1$ will serve as the source to estimate $\mu_t^x$ in eq.37.

## F.3 INPAINTING BASED GENERATION

For stroke based generation, we provide two types of conditional generation.

## G    ABLATION STUDY OF STOKE-BASED CONDITIONAL GENERATION

In order to investigate the diversity and faithfulness of stoke-based conditional generation, we conduct the ablation study with respect to the hyperparameter $\xi$.

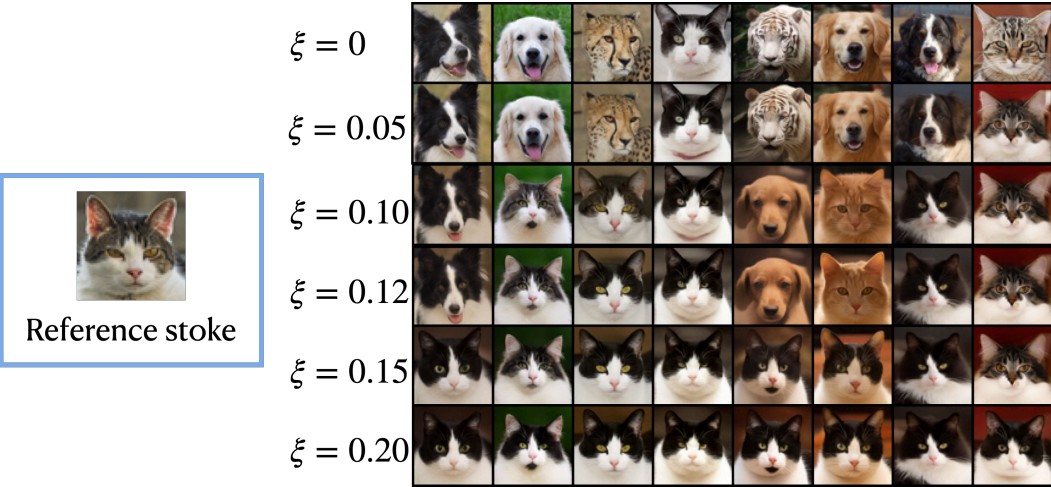

Figure 7: Ablation study for the stoke-based conditional generation. When $\xi = 0$, it is unconditional generation.Notably, the diversity of the generation will decay when we increase $\xi$. In order to achieve a balance between faithfulness and diversity, one needs to tune the hyperparameter $\xi$.

## H    ADDITIONAL FIGURES

We demonstrate the samples for different datasets with varying NFE.

### H.1    TOY DATASET COMPARED WITH CLD

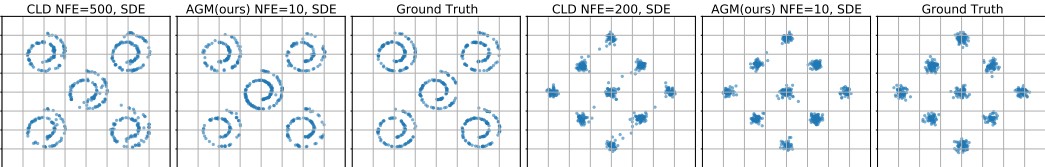

Figure 8: The comparison with CLD(Dockhorn et al., 2021) using same network and stochastic sampler SSS, for Multi-Swiss-Roll and Mixture of Gaussian datasets. We achieve visually better results with one order less NFEs.

## H.2   AFHQv2 Inpainting Generation

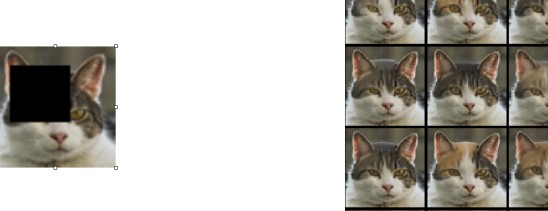

**Conditional Data**                    **Generated Data**

Figure 9: AGM-ODE Uncured inpainting generation

## H.3   AFHQv2 Stroke Based Generation

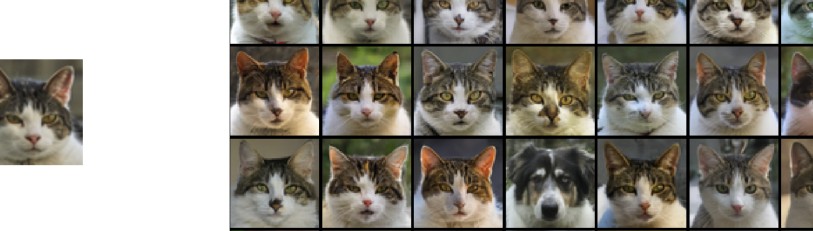

**Conditional Data**                    **Generated Data**

Figure 10: AGM-ODE Uncured stroke based generation

## H.4 CIFAR-10

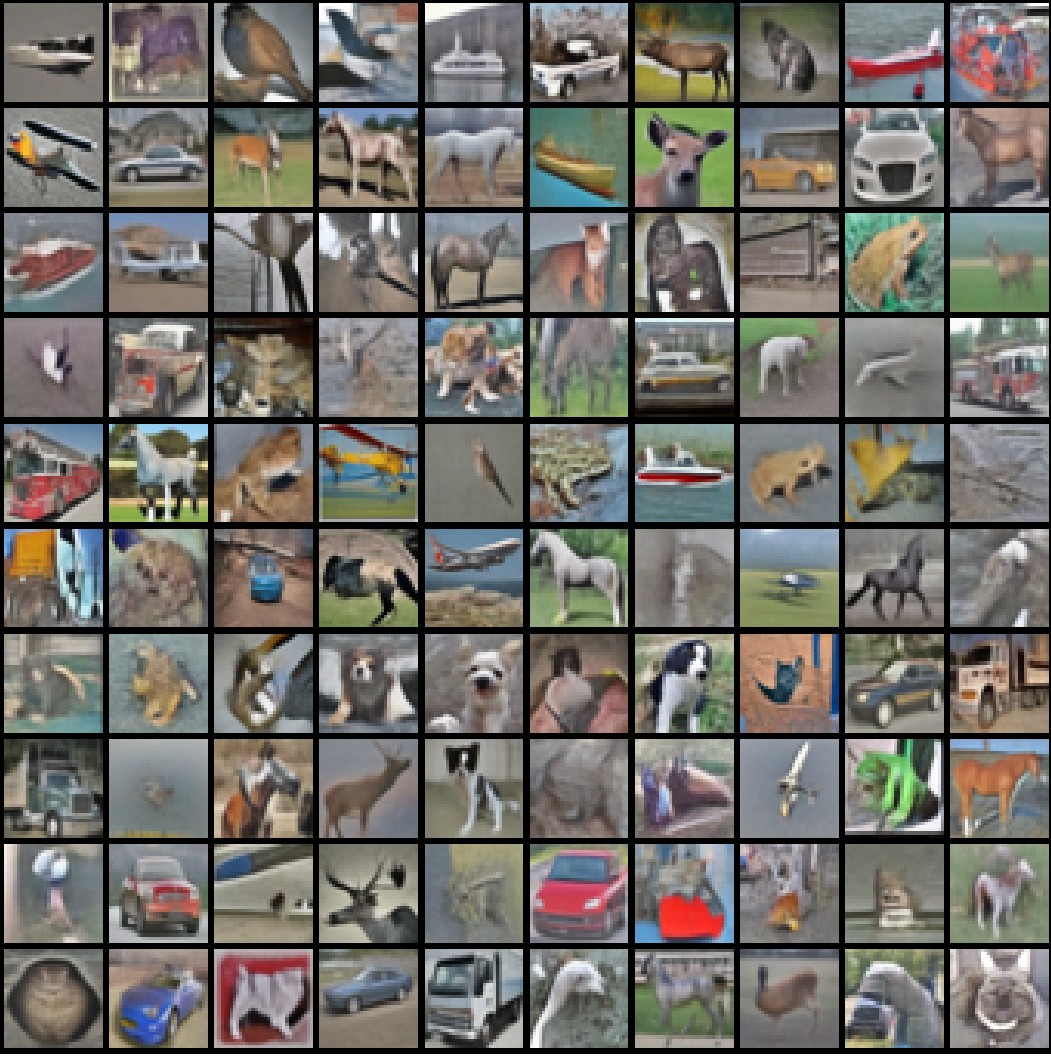

Figure 11: AGM-ODE Uncurated CIFAR-10 samples with NFE=5

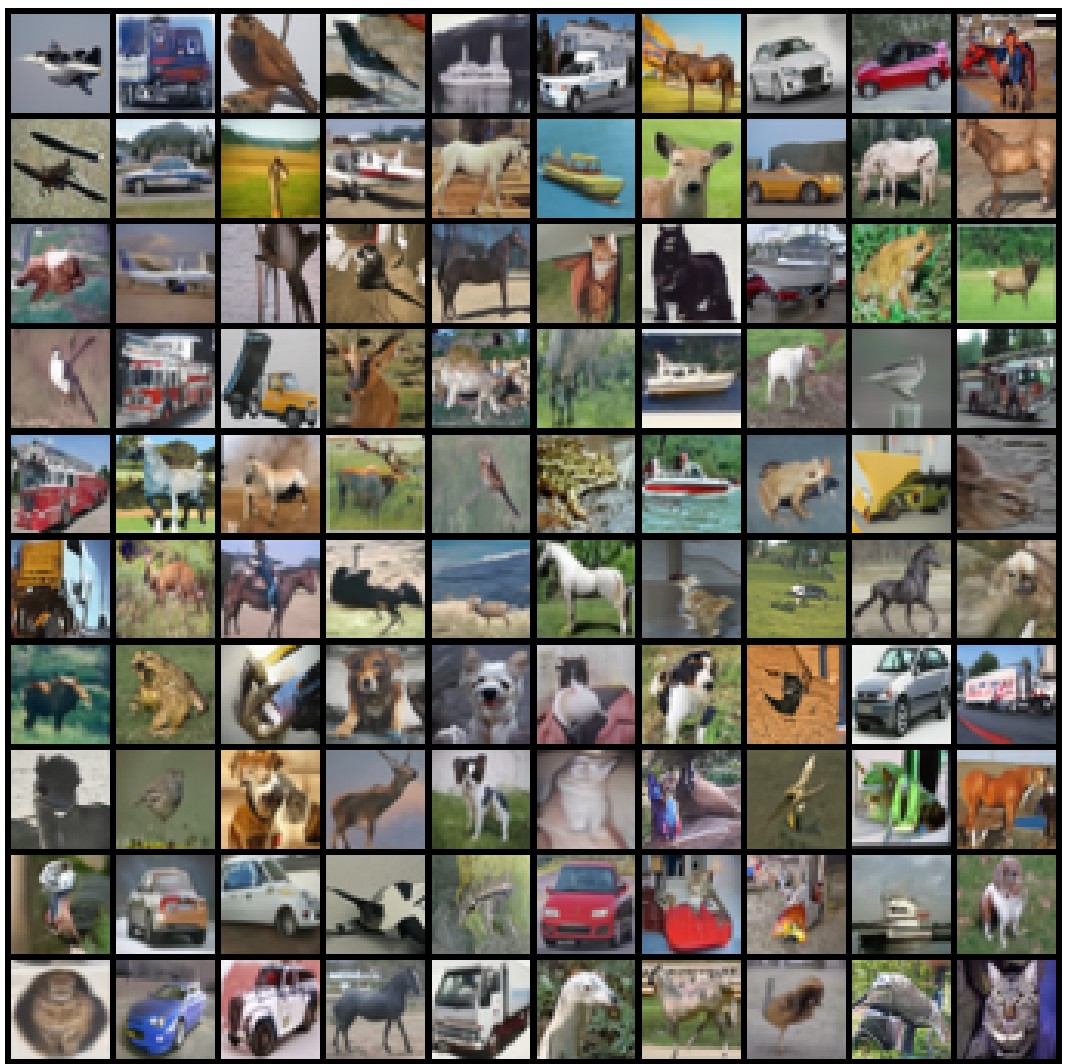

Figure 12: AGM-ODE Uncurated CIFAR-10 samples with NFE=10

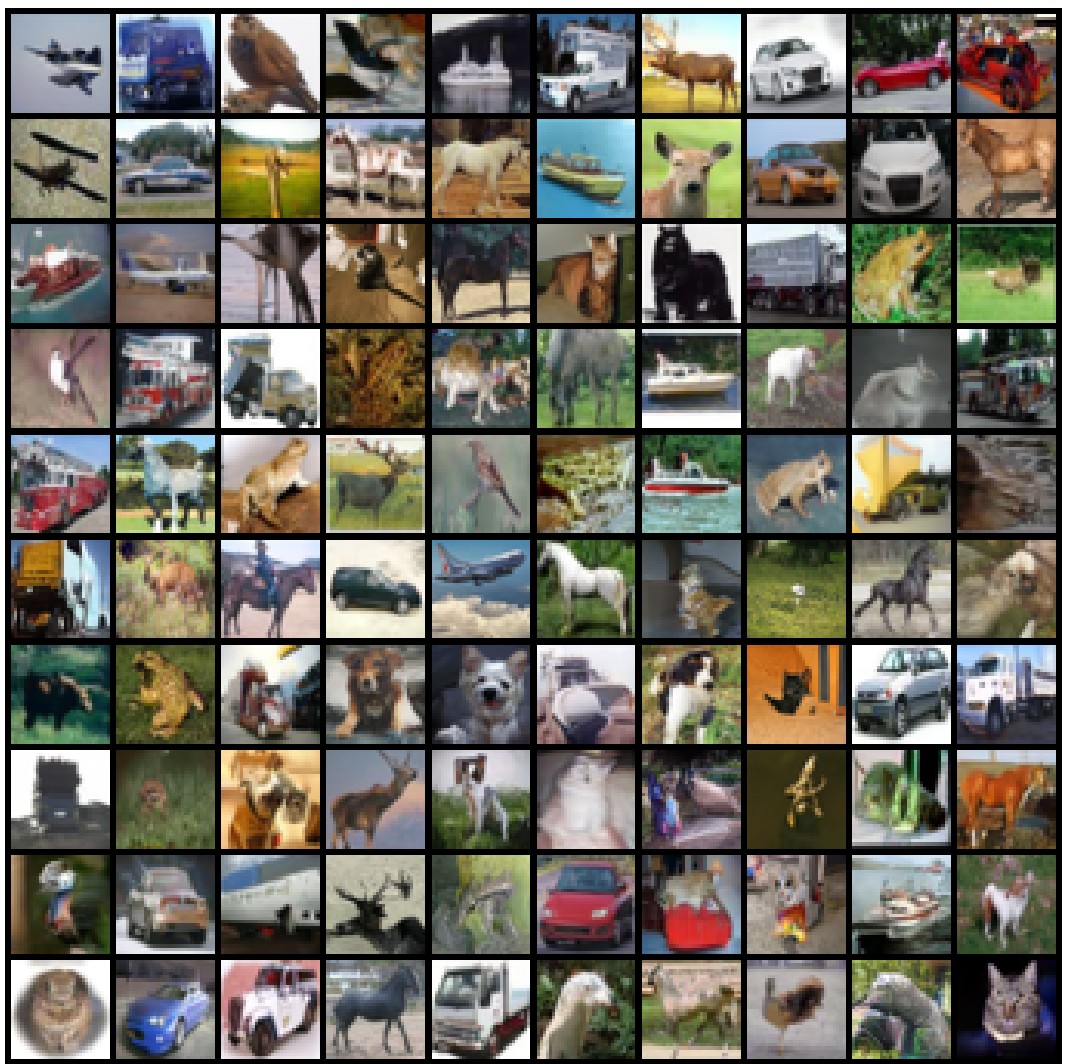

Figure 13: AGM-ODE Uncurated CIFAR-10 samples with NFE=20

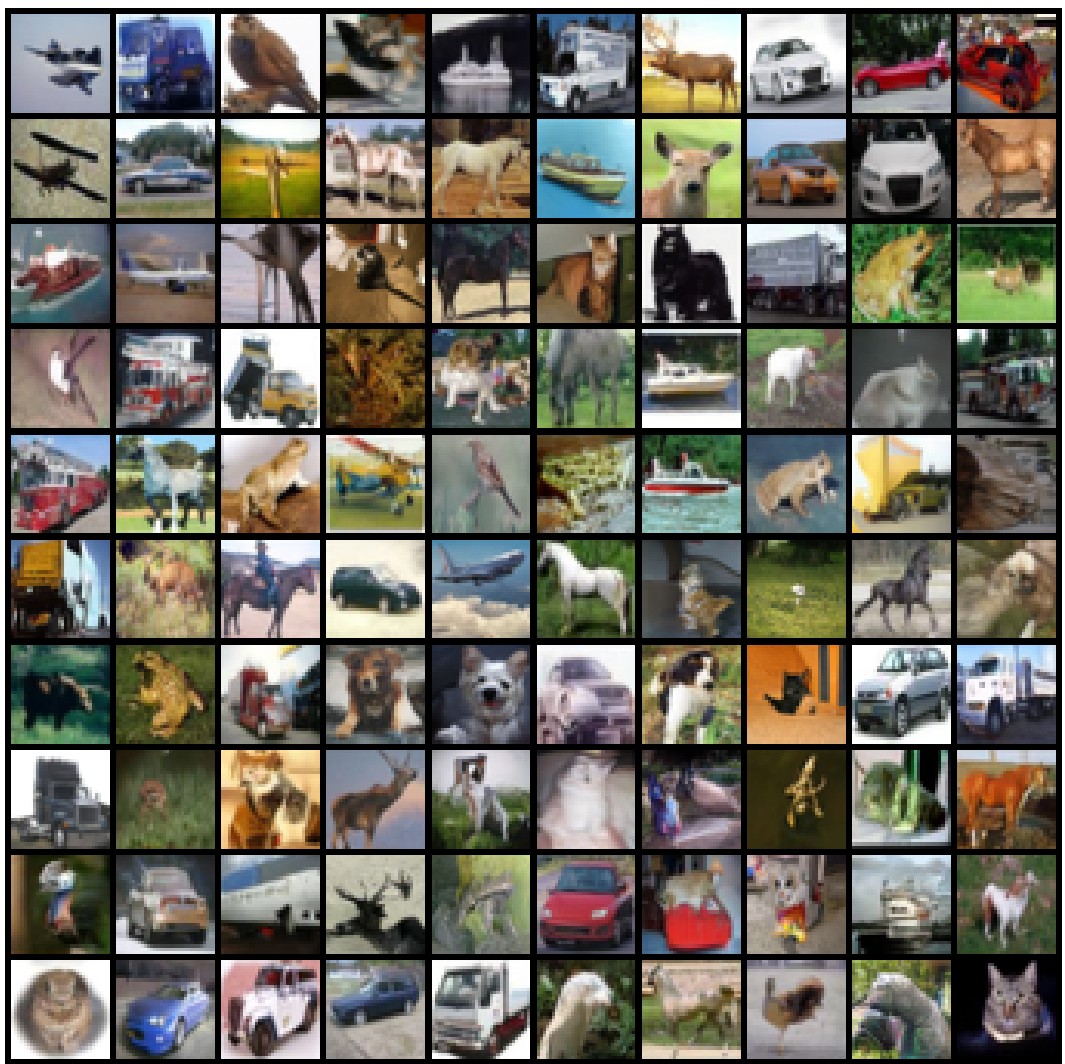

Figure 14: AGM-ODE Uncurated CIFAR-10 samples with NFE=50

## H.5  AFHQv2

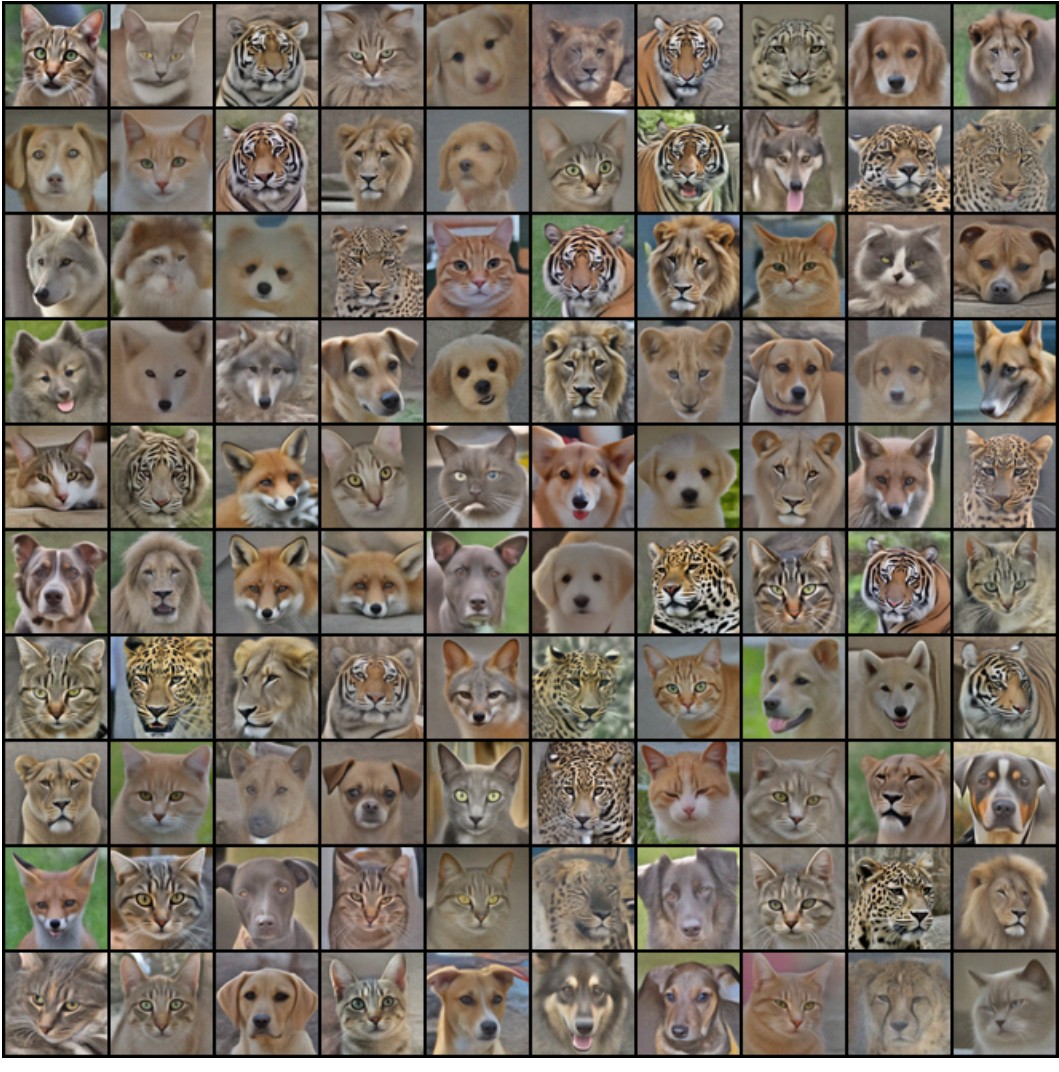

Figure 15: AGM-ODE Uncurated AFHQv2 samples with NFE=5

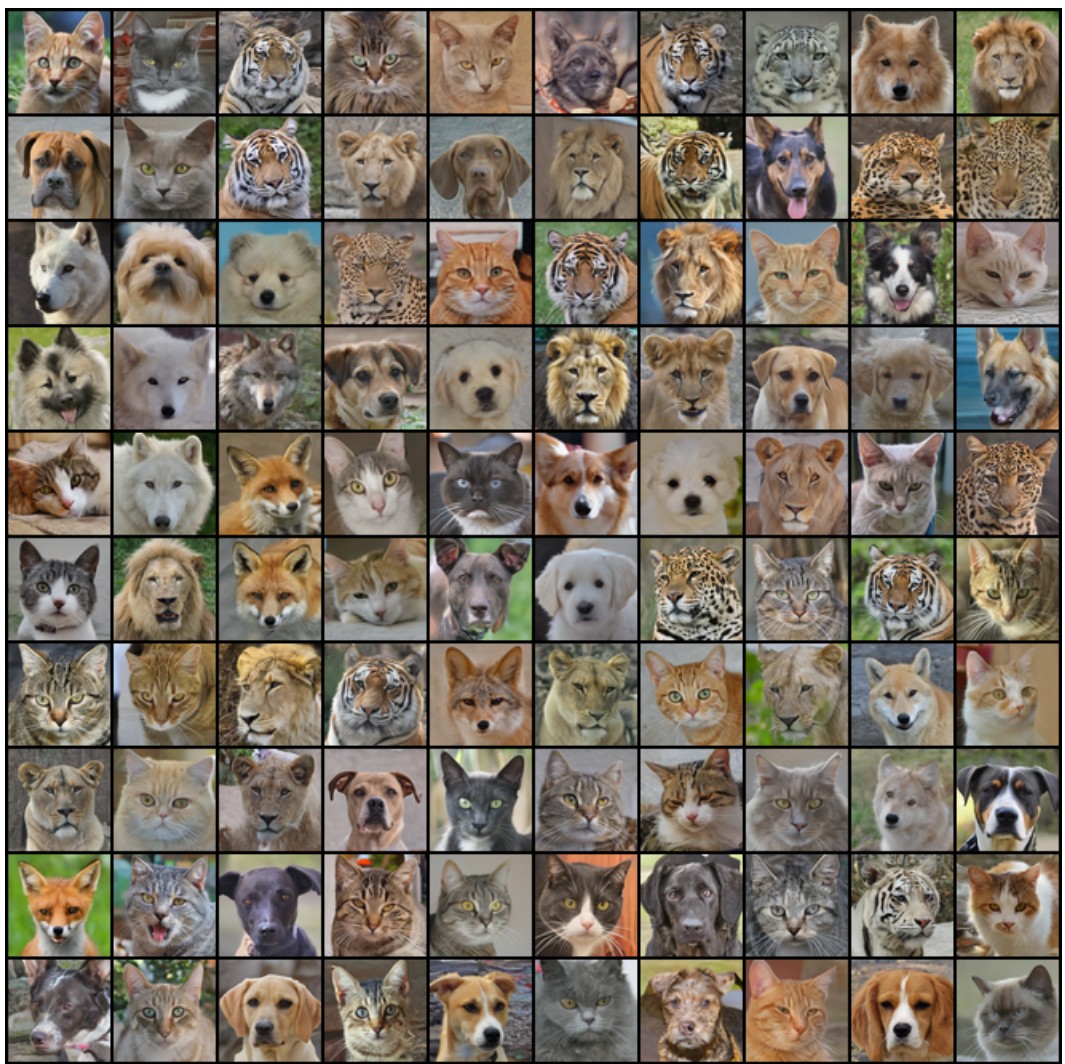

Figure 16: AGM-ODE Uncurated AFHQv2 samples with NFE=10

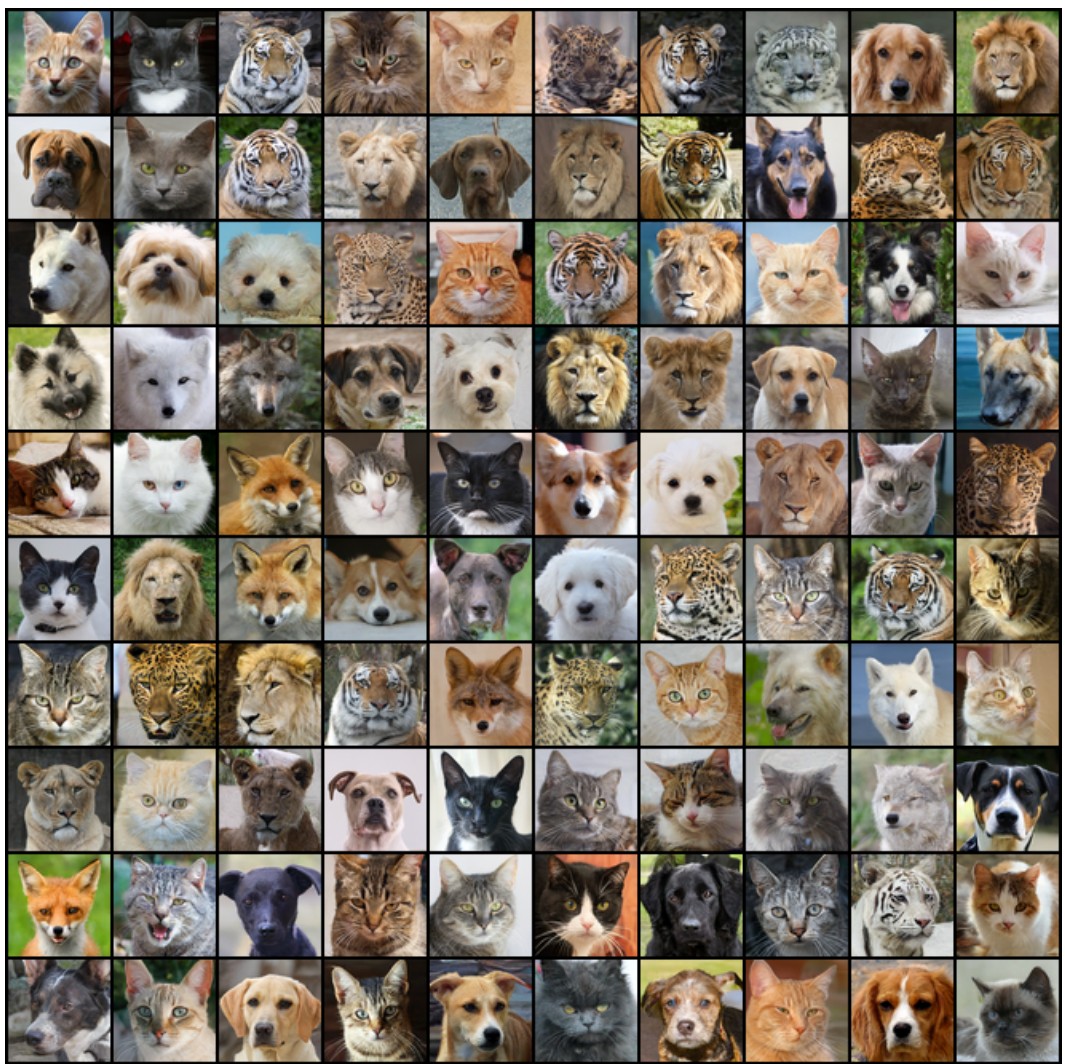

Figure 17: AGM-ODE Uncurated AFHQv2 samples with NFE=20

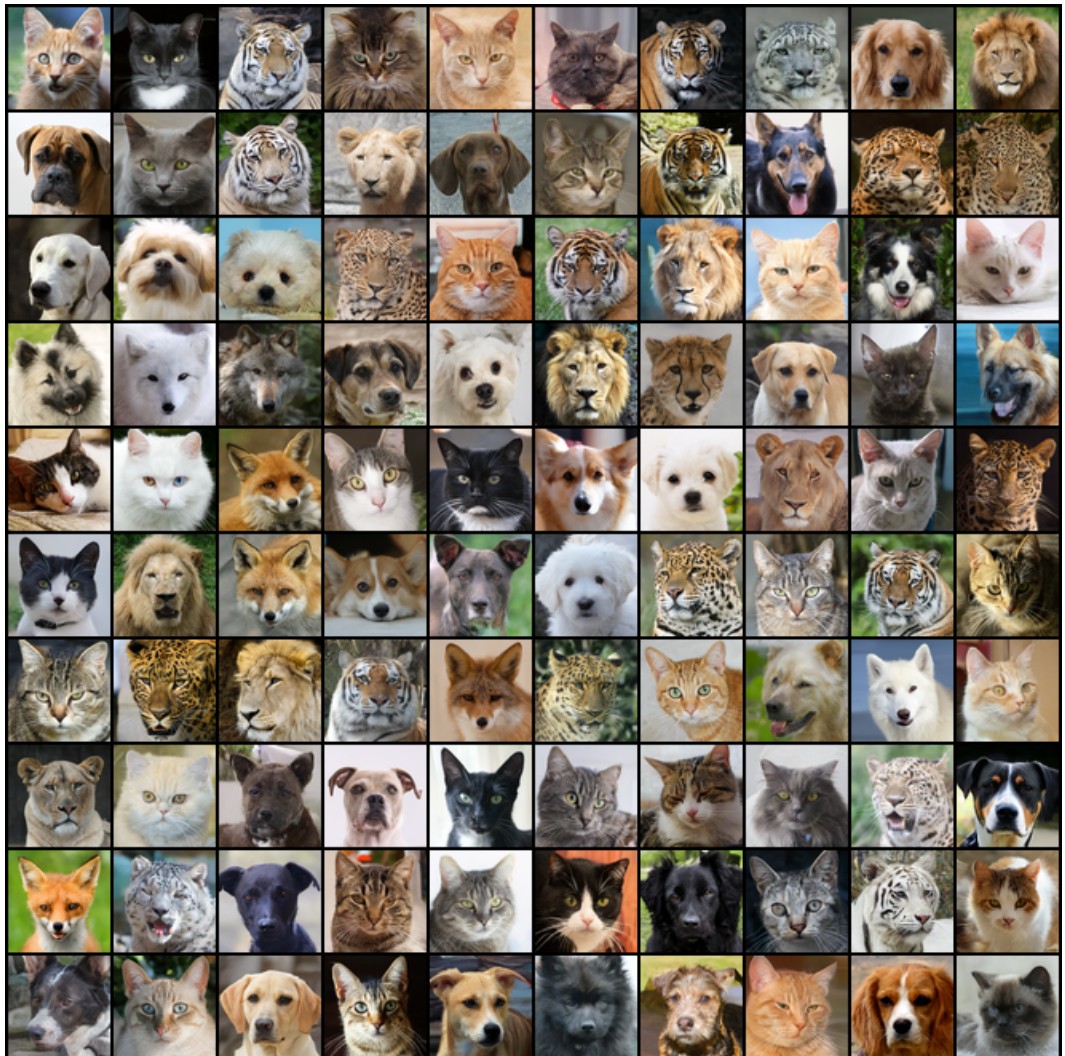

Figure 18: AGM-ODE Uncurated AFHQv2 samples with NFE=50

## H.6 IMAGENET-64

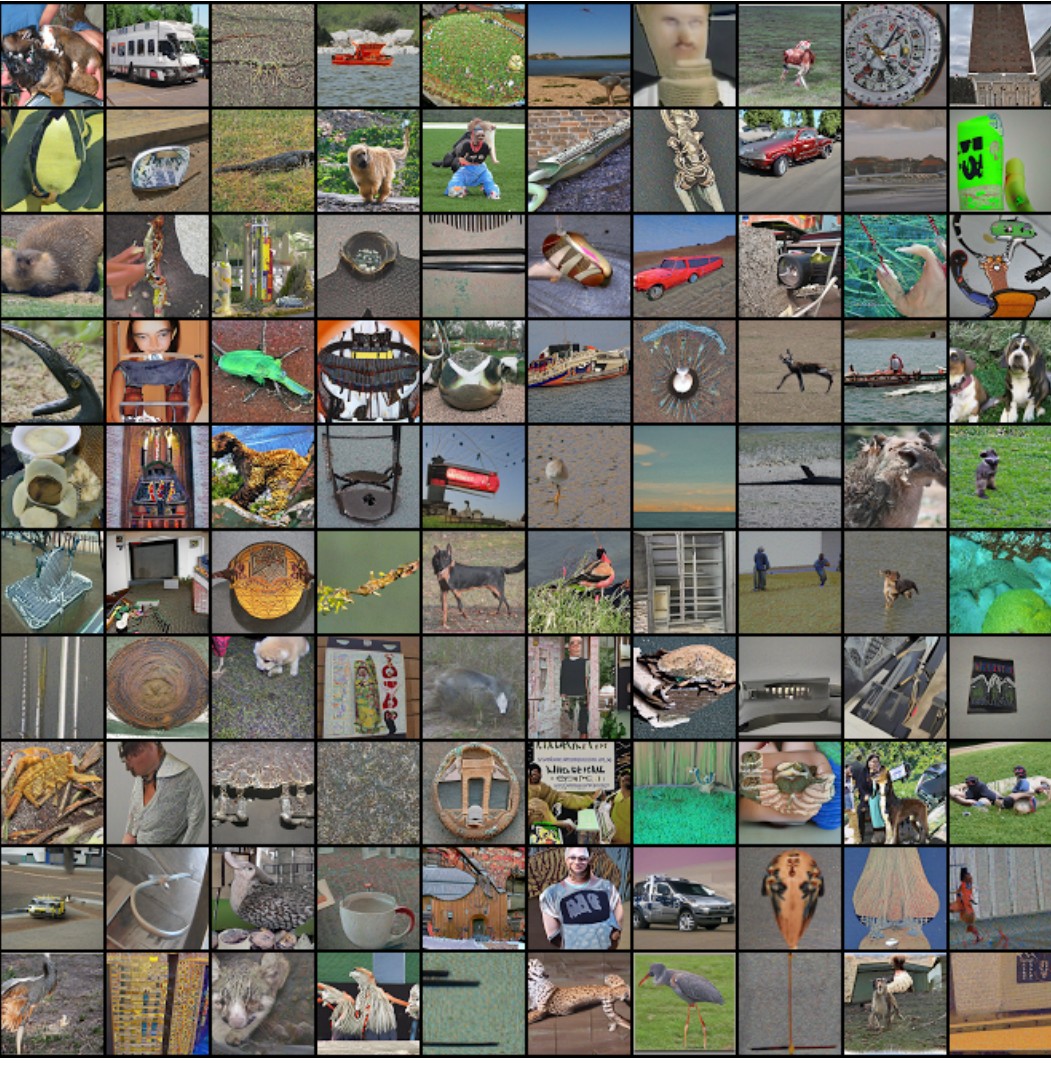

Figure 19: AGM-ODE Uncurated Imagenet-64 samples with NFE=10

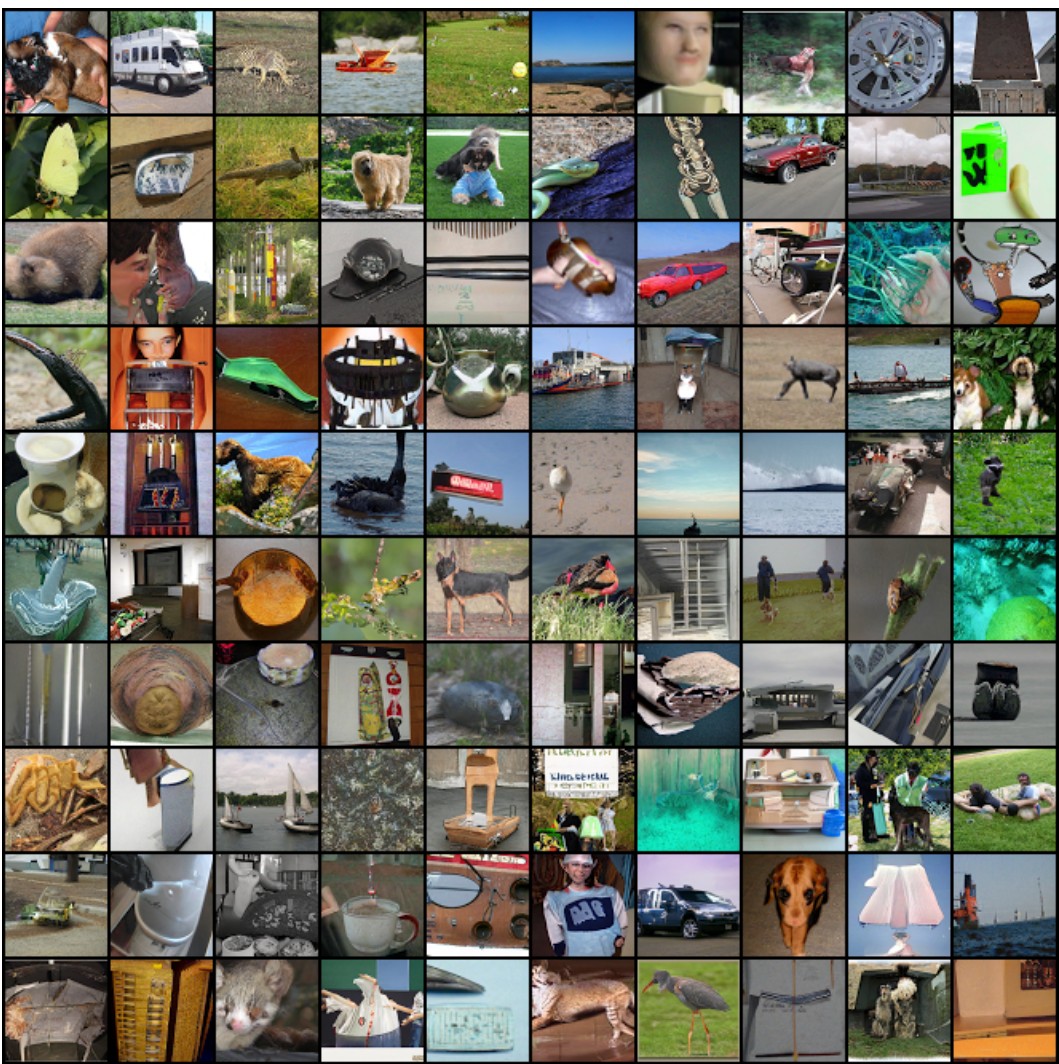

Figure 20: AGM-ODE Uncurated Imagenet-64 samples with NFE=20

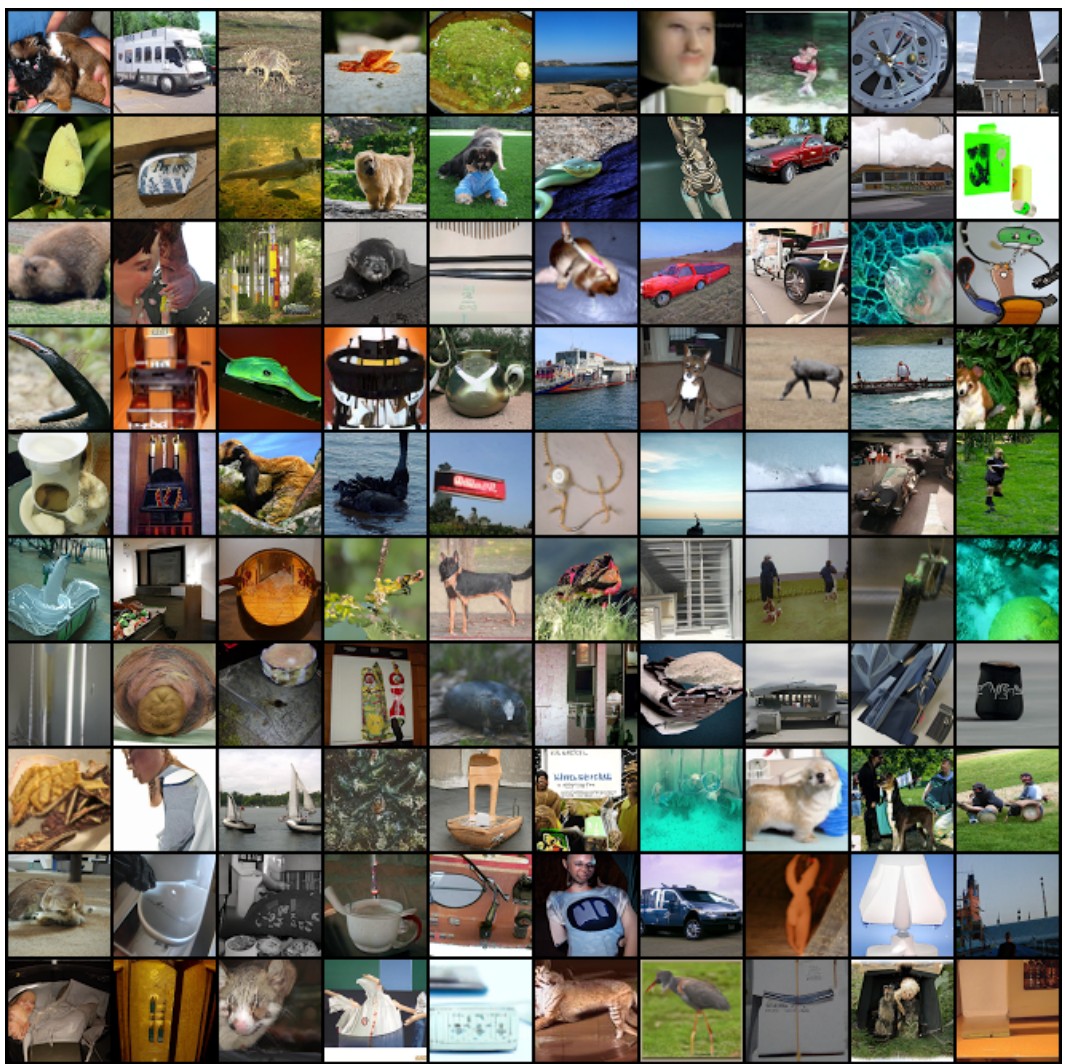

Figure 21: AGM-ODE Uncurated Imagenet-64 samples with NFE=50

