{\mathrm{d}\mathbf{x}_t}{\mathrm{d}t} = \frac{\partial H^*}{\partial \gamma} = \gamma$$
$$\frac{\mathrm{d}\gamma}{\mathrm{d}t} = \frac{\partial H^*}{\partial \mathbf{x}} = 0$$
$$\text{where} \quad \mathbf{x}_0 = x_0 \quad \text{and} \quad \gamma_1 = -\mathbf{r} \cdot (\mathbf{x}_1 - x_1)$$

One can notice that the solution for $\gamma_t$ is the constant $\gamma_t = \gamma = -\mathbf{r} \cdot (\mathbf{x}_1 - x_1)$, hence the solution for $\mathbf{x}_t$ is $\mathbf{x}_t = \mathbf{x}_1 + \gamma t$.

$$\gamma = -\mathbf{r}(\mathbf{x}_1 - x_1) = -\mathbf{r}(\mathbf{x}_0 + (1 - t_0)\gamma - x_1)$$
$$\rightarrow \quad \mathbf{v}_t^* := \gamma = \frac{\mathbf{r}(x_1 - \mathbf{x}_0)}{1 + \mathbf{r}(1 - t_0)}$$

When $\mathbf{r} \to +\infty$, we arrive the optimal control as $\mathbf{v}_t^* = \frac{x_1 - \mathbf{x}_0}{1 - t_0}$. Due to certainty equivalence, this is also the optimal control law for

$$\mathrm{d}\mathbf{x}_t = \mathbf{v}_t \mathrm{d}t + \mathrm{d}\mathbf{w}_t$$

By plugging it back into the dynamics, we obtain the well-known Brownian Bridge:

$$\mathrm{d}\mathbf{x}_t = \frac{x_1 - \mathbf{x}_t}{1 - t}\mathrm{d}t + g_t\mathrm{d}\mathbf{w}_t$$

## C.2 PROOF OF PROPOSITION.3

**Lemma 6.** *The solution of following Lyapunov equation,*

$$\dot{\mathbf{P}} = \mathbf{A}\mathbf{P} + \mathbf{P}\mathbf{A}^{\mathsf{T}} - \boldsymbol{g}\boldsymbol{g}^T \tag{12}$$

*with terminal condition*

$$\mathbf{P}_T = \begin{bmatrix} 0 & 0 \\ 0 & R \end{bmatrix} \tag{13}$$

*is given by*

$$\mathbf{P}_t = \begin{bmatrix} r(t-1)^2 - \frac{1}{3}g^2(t-1)^3 & r(t-1) - \frac{1}{2}g^2(t-1)^2 \\ r(t-1) - \frac{1}{2}g^2(t-1)^2 & g^2(1-t) + r \end{bmatrix}$$

*and the inverse of $\mathbf{P}_t$ is,*

$$\mathbf{P}_t^{-1} = \frac{1}{g^2(-4r + g^2(t-1))(t-1)} \begin{bmatrix} \frac{12(r - g^2(t-1))}{(t-1)^2} & \frac{6(-2r + g^2(t-1))}{t-1} \\ \frac{6(-2r + g^2(-1+t))}{t-1} & 12r - 4g^2(t-1) \end{bmatrix}$$

*When $\mathbf{A} = \begin{bmatrix} 0 & 1 \\ 0 & 0 \end{bmatrix}, \boldsymbol{g} = \begin{bmatrix} 0 \\ g \end{bmatrix}$ and $R = r\mathbf{I}$.*

*Thus,*

$$P_{10} = \frac{-12r + 6g^2(t-1)}{g^2[-4r + g^2(t-1)](t-1)^2} = \frac{-12r}{g^2[-4r + g^2(t-1)](t-1)^2} + \frac{6}{[-4r + g^2(t-1)](t-1)}$$

$$P_{11} = \frac{12r - 4g^2(t-1)}{g^2[-4r + g^2(t-1)](t-1)} = \frac{12r}{g^2[-4r + g^2(t-1)](t-1)} + \frac{-4}{[-4r + g^2(t-1)]}$$

*Proof.* One can plug in the solution of $\mathbf{P}_t$ into the Lyapunov equation $\mathbf{P}_t$ and it validates $\mathbf{P}_t$ is indeed the solution.

**Remark 7.** *Here we provide a general form when the terminal condition of the Lyapunov function is not a zero matrix. It explicitly means that it allows that the velocity does not necessarily need to converge to the exact predefined $\mathbf{v}_1$. It will have the same results as shown in the paper by setting $r = 0$.*

$\square$

**Lemma 8.** *The state transition function $\Phi(t, s)$ of following dynamics,*

$$\mathrm{d}\mathbf{m}_t = \begin{bmatrix} 0 & 1 \\ 0 & 0 \end{bmatrix}\mathbf{m}_t\mathrm{d}t$$

*is,*

$$\Phi(t, s) = \begin{bmatrix} 1 & t - s \\ 0 & 1 \end{bmatrix}$$

*Proof.* One can easily verify that such $\Phi$ satisfies $\partial\Phi/\partial t = \begin{bmatrix} 0 & 1 \\ 0 & 0 \end{bmatrix}\Phi$. $\square$

**Lemma 9** (Chen & Georgiou (2015)). *The optimal control $\mathbf{u}_t^*$ of following problem,*

$$\mathbf{u}_t^* \in \arg\min_{\mathbf{u}_t \in \mathcal{U}} \mathbb{E}\left[\int_0^T \frac{1}{2}\|\mathbf{u}_t\|^2\right]dt$$

$$s.t \quad d\mathbf{m}_t = \begin{bmatrix} 0 & 1 \\ 0 & 0 \end{bmatrix}\mathbf{m}_t dt + \mathbf{u}_t dt + \mathbf{g}d\mathbf{w}_t$$

$$\mathbf{m}_0 = m_0, \quad \mathbf{m}_1 = m_1$$

*is given by*

$$\mathbf{u}_t^* = -\mathbf{g}\mathbf{g}^\mathsf{T}\mathbf{P}_t^{-1}\left(\mathbf{m}_t - \Phi(t,1)\mathbf{m}_1\right)$$