# OpenReview forum: "Generative Modeling with Phase Stochastic Bridge"
_ICLR.cc/2024/Conference — ICLR 2024 oral_

### Official Review · Reviewer_5NhC · 2023-10-25

**Soundness:** 4 excellent
**Presentation:** 3 good
**Contribution:** 4 excellent
**Rating:** 8
**Confidence:** 2

**Summary:**

The paper presents a novel generative modeling framework called Acceleration Generative Modeling (AGM), which is grounded in phase space dynamics. The authors leverage insights from Stochastic Optimal Control to construct a path measure in the phase space that enables efficient sampling. The framework demonstrates the capability to generate realistic data points at an early stage of dynamics propagation, which sets the stage for efficient data generation by leveraging additional velocity information along the trajectory. The model yields favorable performance over baselines in the regime of small Number of Function Evaluations (NFEs) and rivals the performance of diffusion models equipped with efficient sampling techniques.

**Strengths:**

1. The proposed AGM framework offers a new perspective on accelerating sampling in generative modeling by leveraging additional velocity information.

2. The model demonstrates competitive results compared to diffusion models equipped with efficient sampling techniques, particularly in small NFE settings.

3. The paper provides a clear and detailed explanation of the AGM framework, its training, and sampling procedures.

**Weaknesses:**

1. The paper could provide more insights into the potential applications of the AGM framework beyond image generation.

2. The paper could discuss potential improvements to the AGM framework, such as enhancing the training quality through data augmentation, fine-tuned noise scheduling, and network preconditioning.

**Questions:**

Can the AGM framework be applied to other domains beyond image generation, such as natural language processing or time series data?

---

> ### Author Response · Authors · 2023-11-16
> **To Reviewer 5NhC**
>
> We deeply thank the reviewer for all the comments. The summary is accurate and the questions are interesting and helpful.
>
> Please kindly see our itemized replies below to address the reviewer’s concerns.
>
> #### **1. potential applications of the AGM framework beyond image generation.**
> This is a great suggestion and we do believe that this framework might be particularly useful for trajectory inference tasks in which the Newtonian dynamics (phase dynamics) constraint is imposed, for example, single cell trajectory inference [2] or molecular dynamics  [3] simulation. Combining our framework with the approach described in reference [1] would not only be interesting but could also provide a scientifically robust foundation for addressing these trajectory inference challenges.
>
> Regrettably, owing to the authors' limited expertise in these particular domains, they cannot make further concrete assertions regarding the appropriateness of AGM within these domains.
>
> [1] Liu, Guan-Horng, et al. "Generalized Schr\" odinger Bridge Matching." arXiv preprint arXiv:2310.02233 (2023).
>
> [2] Tianrong Chan et al . 'Deep Multi Marginal Momentum Schrodinger Bridge'
>
> [3] Holdijk, Lars, et al. "Stochastic Optimal Control for Collective Variable Free Sampling of Molecular Transition Paths." Thirty-seventh Conference on Neural Information Processing Systems. 2023.
>
> #### **1. Can the AGM framework be applied to other domains beyond image generation, such as natural language processing or time series data?**
> Similar to the diffusion model [1,2] for discretized variables in language generation tasks, it is possible to utilize AGM for language generation. In this case, the transition matrix $\Phi(\cdot,\cdot)$ will be matrix-lized. We leave further explorations in sequence modeling as future work which we believe is an exciting direction.
>
> [1]Zhang, Yizhe, et al. "PLANNER: Generating Diversified Paragraph via Latent Language Diffusion Model." arXiv preprint arXiv:2306.02531 (2023).
>
> [2] Austin, Jacob, et al. "Structured denoising diffusion models in discrete state-spaces." Ad

---

### Official Review · Reviewer_76FN · 2023-10-27

**Soundness:** 3 good
**Presentation:** 3 good
**Contribution:** 2 fair
**Rating:** 8
**Confidence:** 4

**Summary:**

The papers uses the tools of stochastic optimal control theory to define the forward pass for a kind of generative diffusion model strictly related to Diffusion Schrödinger Bridge Matching. The approach combines the velocity augmentation used in Critical-damped Langevin Dynamics with the bridge approach by solving a linear Gaussian control problem in closed-form. This solution leads to relatively straight paths that are suitable for fast-sampler acceleration both in the stochastic and in the deterministic case. The method has competitive performance for small numbers of functional evaluations, but it lags behind other methods when more evaluations are used (>100).

**Strengths:**

- The paper uses stochastic control theory effectively in order to construct a forward process with the desired properties. I believe that this is highly promising as optimal control and diffusion modeling are deeply related and many advanced control techniques can be imported in the diffusion literature using a similar approach.

- The paper provides a rigorous description of the algorithm and the math behind it without requiring an excessive level of mathematical sophistication in the reader.

- The experiments are rigorous and comprehensive and properly show the performance profile of the method and most relevant baselines under different conditions.

**Weaknesses:**

- The exposition is rather dense and, as a consequence, the paper is somewhat difficult to read. This is a pity since the underlying concept are rather intuitive and can be understood by a wide audience.

- As also stated by the authors, the performance of the method is inferior to several baselines for a large number of functional evaluations. However, I do not think that this is a major issue since this class of models are generally designed to work well in the low NFE range, and the results are good in this relevant range. It is quite intuitive to me that there should be a trade off between straight paths and high NFE performance, since the smoothness constraints can limit the expressivity and probabilistic coverage of the method.

**Questions:**

I find the pseudocode in Algorithm 1 and 2 to be rather uninformative. A good pseudo-code should allow the reader to implement the algorithm almost without referring to the rest of the paper. In this case, the most important parts of the code (e.g. the form of the loss) are omitted. Could you update it to make it more self-contained?

- The idea of the initial velocity conditioning  is interesting, but it is difficult to evaluate its potential without quantitative results and comparisons. Intuitively, it seems to me that it will likely lead to a substantial drop in diversity. Can you report the FID for the conditional sampler?

---

> ### Author Response · Authors · 2023-11-16
> **To Reviewer 76FN**
>
> We extend our sincere gratitude to the reviewer for the valuable comments. Please kindly find below our itemized responses, presented in an effort to address each of the reviewer’s concerns.
>
> #### **1. The exposition is rather dense and, as a consequence, the paper is somewhat difficult to read.**
> We apologize for the difficulties caused by the dense exposition. We add one more gentle introduction of SOC in the appendix which hopefully can increase the readability.
>
> #### **2. pseudocode in Algorithm 1 and 2 to be rather uninformative.**
> Thanks for this valuable suggestion. In the revised version, we have included an enhanced pseudocode that provides more comprehensive information.
>
> #### **3. For conditional generation, it seems to me that it will likely lead to a substantial drop in diversity.**
> The velocity conditioning experiment was designed to qualitatively showcase the properties of the velocity space, but the reviewer touched upon a subtle but very interesting complication (thank you!). It is indeed leading to a substantial drop in diversity. We found that it corresponds to the value of the hyperparameter $\xi$. We provide an ablation study in the Appendix F. When $\xi=0$, the trajectory degenerates to the unconditional case which leads to the highest diversity with the lowest faithfulness of reference data point. When $\xi$ is large, the diversity drops dramatically but the faithfulness increases. To achieve a balance between faithfulness and diversity, one needs to tune the hyperparameter $\xi$.

---

> > ### Comment · Reviewer_76FN · 2023-11-21
> >
> > Dear authors,
> >
> > I am satisfied by the revision and I am happy that my observations were useful. As reflected in my original score, I am strongly in favor of accepting this work.

---

### Official Review · Reviewer_rgEV · 2023-11-01

**Soundness:** 3 good
**Presentation:** 4 excellent
**Contribution:** 4 excellent
**Rating:** 8
**Confidence:** 3

**Summary:**

The work proposes Acceleration Generation Modeling (AGM) as an extension to Critical-damped Langevin Dynamics (CLD) based on the theoretical results of stochastic optimal control. The proposed acceleration term has the effect of straightening the sample trajectories in the sampling process and reducing sampling complexity. The linearity of sampling trajectory enables the AGM generation process to take less number of evaluations and make sampling hops. AGM is compatible with both deterministic (ODE) and stochastic (SDE) samplers. Experiment results on CIFAR-10, AFHQ, and ImageNet show that AGM demonstrate competitive results with less number of evaluations with smaller number of evaluations.

**Strengths:**

1. Overall speaking, the proposed idea is simple yet effective and the motivation is backed by solid theoretical results in the domain of stochastic optimal control.
2. Both quantitative and qualitative results support the motivation of AGM. Quantitative results under different settings show AGM is able to achieve competitive or better results with similar to less number of evaluations. Qualitative results also show the better ability of AGM to make sampling hops and recover the denoised images at an early stage compared to CLD.
3. The presentation of the work is also of high quality. The introduction of the theoretical results is concise but also critical to motivate the proposal of AGM. The rest of Section 3 presenting AGM in technical details is also well-structured and easy to follow.

**Weaknesses:**

The work does not have significant weakness. Minor weakness points include
1. The work only shows experiment results on CIFAR-10, ImageNet 64, and AFHQv2 without scaling to higher resolution images.
2. As the author points out in limitations, AGM is not performing as good as some existing methods especially when the number of evaluations is large. I don't think this is a major weakness as the major benefit of AGM and straight sampling trajectories is the reduced number of evaluations during sampling.

**Questions:**

In Table 4 which shows experiment results on ImageNet 64, DDPM uses a stochastic sample while the other approaches including FM-OT, MFM, and AGM-ODE all use deterministic samplers. This may not be a fair comparison because even for the same type of diffusion model, the sample quality and sampling efficiency could be very different with different types of sampler and deterministic samplers based on ODE numerical methods generally take less number of steps than stochastic sampler. I would suggest the author include both AGM-SDE and AGM-ODE results under different number of evaluations.

---

> ### Author Response · Authors · 2023-11-16
> **To reviewer rgEV**
>
> We deeply thank the reviewer for all the comments. The summary is accurate and the questions are interesting and helpful.
>
> Please kindly see our itemized replies below to address the reviewer’s concerns.
> #### **1. Larger scale dataset**
> We understand the reviewer's concern and we agree that AGM may face unexpected difficulties when scaled to larger resolutions which is also the known issue [1,2] posed for general Dynamical Generative Modeling including Diffusion Model and Flow Matching.  This issue partially (informally) explains the success of latent-space-based Diffusion Models [3,4] in high-resolution scenarios.
>
> Due to the computation resource and time limitation, we were unable to conduct experiments at larger scale about which we apologize.
>
> This is something of high priority in our todo list for further work.
>
> [1] Jiatao Gu et al. 'Matryoshka Diffusion Models'
>
> [2] Zahra Kadkhodaie et al. 'Learning multi-scale local conditional probability models of images.'
>
> [3] Rombach, Robin, et al. "High-resolution image synthesis with latent diffusion models. 2022 IEEE." CVF Conference on Computer Vision and Pattern Recognition (CVPR). 2021.
>
> [4] Vahdat, Arash, Karsten Kreis, and Jan Kautz. "Score-based generative modeling in latent space." Advances in Neural Information Processing Systems 34 (2021): 11287-11302.
>
> #### **1. Unfair comparison with DDPM and FID evaluation for different NFE**
> We totally agree with the reviewer's suggestion and we have removed DDPM from the table.
>
> For AGM, We reported the performance of our model at 0.8M training iteration which turns out the model is under-trained. After the submission, we kept training the model for another 0.8 iterations and now the performance has increased dramatically. We achieved FID 10.97 at 40 NFE compared with SoTA model [1] (to the best of our knowledge) which achieved 11.82 at 132 NFE. We updated the table with varying NFE as reviewer suggested.
>
> Regrettably, in our model, one can only opt to train either AGM-SDE or AGM-ODE, and it is not possible to switch between the two interchangeably like a diffusion model. We acknowledge that this limitation is one of the drawbacks of our approach. Moreover, we deeply apologize for not being able to train an AGM-SDE from scratch during the rebuttal phase, primarily due to the significant training complexity associated with ImageNet. Given our current computational resources (8 x Nvidia A100), training an AGM-SDE from scratch would require approximately 8 weeks [1], for which we sincerely apologize.
>
> [1] Karras et al. "EDM"

---

### Official Review · Reviewer_gcJ1 · 2023-11-07

**Soundness:** 3 good
**Presentation:** 3 good
**Contribution:** 3 good
**Rating:** 8
**Confidence:** 4

**Summary:**

This paper proposed the acceleration generative model (AGM), which is Bridge Matching method with a dynamics-based diffusion model with stochastic optimal control (SOC) theory that rectifies the trajectory of the second-order momentum dynamics which is first introduced in CLD. First, the optimal acceleration function is derived to the solution of the stochastic bridge problem, which is given by minimizing the SOC objective function. Different from CLD whose velocity field is defined by the score function of the pre-defined critically-damped Langevin diffusion process, the velocity field of AGM is learned to rectify the particle trajectory. This SOC problem is designed to minimize the distance between the ground-truth (GT) destination and the trajectory destination. Then like in CLD, this paper took advantage of the momentum-based approaches and proposed that the sampling-hop, the estimated data point $x_1$ given the early sampling stage outputs, is predicted more accurately compared to existing methods. In the empirical experiments, the sampling quality is improved especially in the low-NFE regime.

**Strengths:**

* The idea of rectifying the particle trajectory with the velocity field is an intuitive approach, which is widely used in the literature. Existing works used handcrafted way to design the velocity field, but this paper aimed to both optimize this part of the stochastic process by using the SOC theory.
* The objective is well-defined: when the control regularization term approaches to zero, then the objective directly turns into the square mean
* Even though the velocity should be trained, the whole training process is simulation-free: we do not need any further simulation process like in current SoTA models that require further self-distillation for high-quality image generation in low-NFE regime. Furthermore, this method can be pipelined with the distillation techniques like other diffusion model methods.

**Weaknesses:**

* The clarity of the paper will be better if the conditions of the Lemmas and Propositions written in this paper is stated more concretely and with full notations, especially in the appendix.
  - In the sampling-hop part, the writing does not fully cover how the sampling-hop is more accurately evaluated compared to the CLD case. Both this method and CLD make predictions of the data from both the current state and the velocity, while the compared EDM (Figure 2) does it from state alone.
  - In the Probabilistic ODE part of (7), an additional notation rather than $g(t)$ is recommended to be used, like $g_t \to h_t$ in the matrix notation and $g(t)=h_t$ for BM-SDE part. Because the notation $g(t)$ or $g_t$ is abused, it can be misleading that the score term of the probabilistic ODE is neglected.
* The SOC theorem is only used limitedly; the regularization in terms of $\int ||a_t||^2$ is ignored and this can threaten the stability of the acceleration space, even though this is not directly revealed in the paper.
* Whereas the theoretical background is sound and the improved performance is guaranteed, the hyperparameters such as the diffusion coefficients and the SDEs are not optimized, which causes its lacking performance compared to EDM (look at Figure 5). However, this is expected to be enlightened with further works.

**Questions:**

* It will be helpful if the acceleration coefficient $a_t$ for image datasets is depicted, as the momentum of how the image data is being generated in Figure 1 or Figure 2. It is expected that the acceleration coefficients show similar semantic features like $x_t$ and $v_t$, but have varying scales.
* Can you provide elementary introduction of the stochastic optimal control? While this paper works only the simple case of the SOC (no regularization case), introducing some details, or at least some introductory materials will help the readers to follow up the backgrounds.
* I guess that the ImageNet64 performance is not yet optimized: the generative performance of the SoTA models are expected to be much better than the paper have proposed. I think at least the performance should be compared with CLD-SGM from the same architecture.

================================

* It will be helpful for the readers' understanding if you use the colored hyperlinks by reference (\ref) or citation (\cite, \citep, \citet) commands.

**Details Of Ethics Concerns:**

None.

---

> ### Author Response · Authors · 2023-11-16
> **To Reviewer gcJ1**
>
> We would like to express our sincere gratitude for your valuable feedback and comments. We truly appreciate the time and effort you invested in assessing our submission.
>
> Please kindly see our itemized replies below in order to address the reviewer’s concerns.
>
> #### **1. The clarity of the paper will be better if the conditions of the Lemmas and Propositions written in this paper is stated more concretely**
> Thank you for raising this issue. we have made revisions to the notations and refined the statements of the lemmas and propositions (Lemma 7 and Section D.10 in appendix, Proposition 5 in main paper).
> #### **2. Both this method and CLD make predictions of the data from both the current state and the velocity, providing the data prediction result from CLD.**
> Thank you for mentioning this aspect which is indeed valuable.
>
> It has come to our attention that the training objective of CLD solely focuses on the score function with respect to velocity (scaled $\epsilon_1$). Consequently, there is no apparent method for CLD to effectively reconstruct the data point using the intermediate state and velocity. Specifically, in order to reconstruct $x_1$,the information of stochasticity in the position channel, denoted as $\epsilon_{0}$, is required. However, it is important to note that this information is not available after the training phase of CLD. Therefore, the reconstruction of $x_1$ is not feasible.
>
> In our case, the special structure of acceleration, which includes the stochasticity information of state and velocity channel (eq.9 in revision), allows us to reconstruct the $x_1$ (Proposition.5).
>
> To address any confusion resulting from our previous insufficient explanation, we have included additional clarification in Section 3.2 in revision.
>
> #### **3. Notation abuse in eq.7.**
> Thank you for careful reading! We have fixed this issue in the revision!
> #### **4. provide an elementary introduction of the stochastic optimal control?.**
> Thanks for the valuable suggestion. We have provided a gentle introduction of stochastic optimal control in the appendix C.
> #### **5. the ImageNet64 performance is not yet optimized,  the generative performance of the SoTA models are expected to be much better than the paper has proposed.**
> The experiment we evaluate on is the **unconditional** imagenet-64 and we do not have computation resources to conduct the training on this dataset for all baselines. As a result, We report the performance of baseline models from a recent paper (see Table.8 from [1]).
>
> For AGM, We reported the performance of our model at 0.8M training iteration which turns out that the model was under-trained. After the submission, we kept training the model for another 0.8M iterations and the performance increased dramatically. We achieved FID 10.97 at 40 NFE compared with SoTA model [1] (to the best of our knowledge) which achieves 11.82 at 132 NFE.
>
> We kindly invite the reviewer to suggest any other baselines we are missing, and we are more than happy to incorporate them into our paper.
>
> [1] Pooladian, Aram-Alexandre, et al. "Multisample flow matching: Straightening flows with minibatch couplings." arXiv preprint arXiv:2304.14772 (2023).
> #### **6. HyperLink color**
> We have added the hyperlink color back in the revision for better visualization as the reviewer suggested!
>
> #### **7. Trajectory of acceleration**
> We have added the plot of position, and velocity together with acceleration in the appendix G in the revision. Upon reviewing Appendix G, it becomes evident that the trajectories of the variables remain consistent over varying NFEs, despite the trajectories of any variable not being perfectly linear in the infinite timestep limit.
> #### **8. The problem formulation ignores the regularization of acceleration**
> We would like to respectfully stress that the regularization of the acceleration $||a_t||_2^2$ is not ignored in our framework (see, eg, Equation 5).
>
> To be more precise, as $r$ approaches positive infinity, there still exist multiple possible stochastic processes which start from $x_0$ at $t=0$ and converge to $x_1$ at $t=1$ and all of them are minimizers of the objective function **without** regularization. Among these processes, our preferred stochastic process should also minimize the control effort $\int_{0}^{1}||a_t||_2^2 dt$ (regularization). This solution, which aligns with the unique solution presented in Definition 2 (In general, the solution of SOC is not unique. But in our simple Linear Quadratic case, the solution is unique, see Theorem 6.1 in [this textbook](https://www.control.utoronto.ca/people/profs/kwong/ece410/2008/notes/chap6-08.pdf)), is what we have employed to construct our methodology.
>
> To address any confusion that may have arisen, we have provided additional clarification in Appendix C.1 in the revision. This supplementary section aims to resolve any ambiguities and enhance the understanding of the subject matter.

---

> > ### Comment · Reviewer_gcJ1 · 2023-11-22
> > **Response to the authors**
> >
> > 1. Thank you for the detailed response, especially for the regularization of acceleration part in the problem formulation. According to my understanding, the acceleration term $\|\|a_t\|\|_2^2$ was ignored since $R$ goes to infinity. If there exists multiple processes that satisfies the condition without regularization, then your explanation makes sense. Specifically, I am greatly impressed on the complement of abundant materials in the
> >
> > 2. We also appreciate you to precisely mention the difference between CLD and AGM, and I think that this part is one of the key aspect that AGM has improved the CLD-SGM.
> >
> > Since all the important issues are resolved and since I considered that the concept of SOC should be an important blueprint for continuous-time generative model, I raise my review score and suggest this paper as the core contribution of the conference.
> >
> > * * *
> >
> > > Minor clarifications
> >
> > * The state and velocity term in Proposition 5 looks to be mis-compiled. It should be corrected in the camera-ready version.

---

### Author Response · Authors · 2023-11-16
**To all reviewers**

We thank all reviewers for their valuable comments. We are excited that the reviewers identified the novelty of our contribution, appreciated our experimental results and acknowledged the significance of our work.


We have provided a revised version of the draft to address concerns raised by the reviewers. In particular, we integrated the clarifications suggested by all the reviews to make the paper easier to understand for audiences, and corrected all the abused notations, together with additional experimental validations suggested by the reviewers.

We notice that AGM-ODE was under-trained and its performance increased dramatically after another 0.8M iterations of training. Namely, AGM-ODE achieves 3 times faster speed even with better performance (40 NFE with 10.92 FID) compared with mini-batch OT Flow matching [2] (132NFE with 11.82 NFE).

A general itemized summary for all reviewers is listed below, and more detailed responses are provided in the corresponding reply for each reviewer.

## Summary of Revision for all Reviewers
- We provide a gentle introduction of Stochastic Optimal Control  in appendix C.
- We corrected all abused notation (see revised eq.7 and Lemma 7).
- We updated the Table 4 with better numerical results.
- We updated comprehensive pseudo code for improving the clarity of our algorithm.
- We added back colored hyperlinks for better visualization.
- We would like to express our sincere apologies for our inability to fulfill the request for conducting any new training task from scratch on Imagenet-64 due to the computation resource limitation. (Generally, [1] requires 2 weeks training with 32x A100 (See Section F.3 in [1]). And we only have 8xA100 for this project, which implies we may need around 8 weeks to see competitive results compared with [1] or SoTAs.)

[1] Karras et al. 'EDM'

[2] Pooladian, Aram-Alexandre, et al. "Multisample flow matching: Straightening flows with minibatch couplings." arXiv preprint arXiv:2304.14772 (2023).

---

### Public Comment · ~Tianrong_Chen1 · 2024-05-12
**Code release**

The code now is available at https://github.com/apple/ml-agm

---

### Meta-Review · Area_Chair_fk64 · 2023-12-11

**Metareview:**

The paper proposes phase stochastic bridge that works in position and velocity space and enables efficient sampling in the regime of small number of function evaluations. Reviewers unanimously agreed on the merits and the potential of the paper.

To the authors please incorporate all reviewers feedback in the final version of the paper.

**Justification For Why Not Higher Score:**

N/A

**Justification For Why Not Lower Score:**

The contribution of the paper is new  and elegant and it allows efficient sampling on par with diffusion models equipped with efficient and complex samplers.

---

### Decision · Program_Chairs · 2024-01-16

Accept (oral)